# Rare coding variants in *CHRNB2* reduce the likelihood of smoking

Veera M. Rajagopal[1], Kyoko Watanabe [1], Joelle Mbatchou [1], Ariane Ayer[1], Peter Quon[2], Deepika Sharma[1], Michael D. Kessler[1], Kavita Praveen[1], Sahar Gelfman[1], Neelroop Parikshak [1], Jacqueline M. Otto[1], Suying Bao[1], Shek Man Chim [2], Elias Pavlopoulos[2], Andreja Avbersek[2], Manav Kapoor [1], Esteban Chen[1], Marcus B. Jones[1], Michelle Leblanc[1], Jonathan Emberson [3,4], Rory Collins[3], Jason Torres [3,4], Pablo Kuri Morales[5,6], Roberto Tapia-Conyer[5], Jesus Alegre [5], Jaime Berumen [5], GHS-REGN DiscovEHR collaboration*, Regeneron Genetics Center*, Alan R. Shuldiner [1], Suganthi Balasubramanian[1], Gonçalo R. Abecasis [1], Hyun M. Kang[1], Jonathan Marchini [1], Eli A. Stahl[1], Eric Jorgenson [1], Robert Sanchez[2], Wolfgang Liedtke[2], Matthew Anderson[2], Michael Cantor [1], David Lederer[2], Aris Baras [1]✉ & Giovanni Coppola [1]✉

Human genetic studies of smoking behavior have been thus far largely limited to common variants. Studying rare coding variants has the potential to identify drug targets. We performed an exome-wide association study of smoking phenotypes in up to 749,459 individuals and discovered a protective association in *CHRNB2*, encoding the β2 subunit of the α4β2 nicotine acetylcholine receptor. Rare predicted loss-of-function and likely deleterious missense variants in *CHRNB2* in aggregate were associated with a 35% decreased odds for smoking heavily (odds ratio (OR) = 0.65, confidence interval (CI) = 0.56–0.76, $P = 1.9 \times 10^{-8}$). An independent common variant association in the protective direction (rs2072659; OR = 0.96; CI = 0.94–0.98; $P = 5.3 \times 10^{-6}$) was also evident, suggesting an allelic series. Our findings in humans align with decades-old experimental observations in mice that β2 loss abolishes nicotine-mediated neuronal responses and attenuates nicotine self-administration. Our genetic discovery will inspire future drug designs targeting *CHRNB2* in the brain for the treatment of nicotine addiction.

Tobacco smoking is one of the greatest hazards to human health, accounting for over 200 million disability-adjusted life years and 7 million deaths each year globally[1]. The currently available first-line smoking-cessation drugs (varenicline and bupropion) were introduced more than 2 decades ago, even before the Human Genome Project was completed and the genomic revolution started[2–4]. Despite their proven efficacy and wide usage[5], smoking remains a global health hazard, warranting advancements in smoking-related drug-discovery efforts that make use of recent innovations in therapeutic design and delivery[6].

Large-scale rare variant association studies have the potential to advance drug discovery[7–10]. Drug designs inspired by naturally occurring genetic variants that protect humans against diseases have

[1]Regeneron Genetics Center, Tarrytown, NY, USA. [2]Regeneron Pharmaceuticals, Inc., Tarrytown, NY, USA. [3]Clinical Trial Service Unit and Epidemiological Studies Unit, Nuffield Department of Population Health, University of Oxford, Oxford, UK. [4]MRC Population Health Research Unit, Nuffield Department of Population Health, University of Oxford, Oxford, UK. [5]Experimental Research Unit from the Faculty of Medicine (UIME), National Autonomous University of Mexico (UNAM), Mexico, Mexico. [6]Instituto Tecnológico y de Estudios Superiores de Monterrey, Monterrey, Mexico. *Lists of authors and their affiliations appear at the end of the paper. ✉e-mail: aris.baras@regeneron.com; giovanni.coppola@regeneron.com

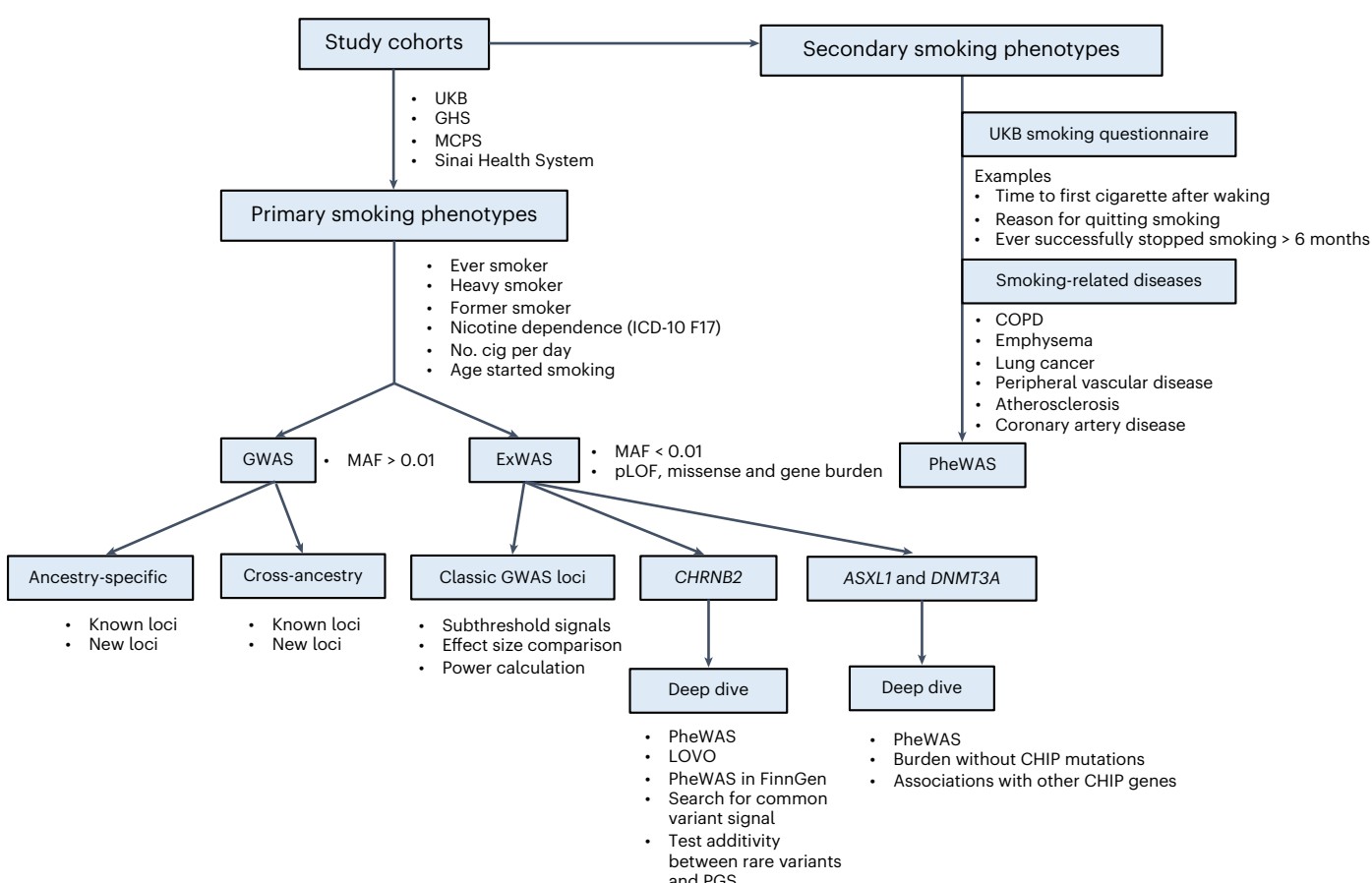

**Fig. 1 | Study design.** The flow chart summarizes the overall study design in terms of cohorts, phenotypes and types of genetic analyses performed. ICD, International Classification of Diseases.

been successful in the past, for example, inhibitors of the enzyme PCSK9 for the treatment of hypercholesterolemia[11–13]. Smoking behavior is strongly influenced by genetics, with twin-based heritability estimates ranging between 45% (for smoking initiation) and 75% (for nicotine dependence)[14]. Genetic variants across the entire minor allele-frequency (MAF) spectrum (common (MAF > 1%), low-frequency (MAF, 0.1–1%) and rare (MAF < 0.1%) variants) contribute to this high heritability[15]. However, human genetic studies of smoking behavior have thus far focused mainly on common and low-frequency variants (that can be imputed with at least moderate accuracy)[16–19]. Such genome-wide association studies (GWASs) were successful in identifying genomic regions associated with smoking. In contrast to GWASs, only a very few rare variant studies of smoking exist to date[15,20]. Although such studies have demonstrated that rare variants contribute substantially to smoking heritability, very few genes have been confidently linked to smoking based on rare variant associations[15,20].

Unlike common variant associations, rare coding variant associations often pinpoint causal genes[21], inform effect direction[21,22], guide follow-up experiments[23] and provide an estimate of the therapeutic efficacy[11,24] and safety[25] of targeting a gene or its product. Even for known drug targets, discovering human genetic evidence is valuable, as it can improve our understanding of the drug mechanisms and help develop new therapeutic modalities to treat diseases[26]. Hence, with the goal of discovering drug targets for smoking, we undertook a large-scale exome-wide association study (ExWAS) of smoking behavior involving up to 749,459 individuals. We studied the associations of rare coding variants in the human genome, captured via exome sequencing, with six major smoking phenotypes and a range of secondary

phenotypes including smoking-related diseases. We also selectively explored the rare variant associations at the known GWAS loci and conducted ancestry-specific and cross-ancestry GWAS meta-analyses for the six smoking phenotypes to validate known loci and identify new loci. Finally, we studied the combined influences of both common and rare variants on smoking behavior.

## Results

### Exome-wide significant associations

The overall study design is shown in Fig. 1. We performed ExWAS meta-analyses for six primary phenotypes (ever smoker, heavy smoker, former smoker, nicotine dependence, cigarettes smoked per day (cig per day) and age started smoking) in sample sizes ranging from 112,670 (cig per day) to 749,459 (ever smoker). The study cohorts and phenotype definitions are described in the Methods, and the cohort-specific sample sizes and participant demographics are summarized in Supplementary Tables 1 and 2, respectively. We focused on coding variants of two functional categories: missense variants and predicted loss-of-function (pLOF) variants (frameshift, splice donor, splice acceptor, stop lost, stop gain and start lost) with MAF < 0.01. In addition to variant-level associations, we also studied gene-level associations, using burden tests in which either pLOF variants only or pLOF and likely deleterious missense variants (that is, predicted to be deleterious by five different algorithms) in a gene are aggregated to create burden masks (or variant sets), which are then tested for association with the phenotypes (Methods)[21]. The burden masks were created using variants at five MAF thresholds (<0.01, <0.001, <0.0001, <0.00001 and singletons) (Supplementary Table 3). Altogether, we performed 8,417,987 association tests across

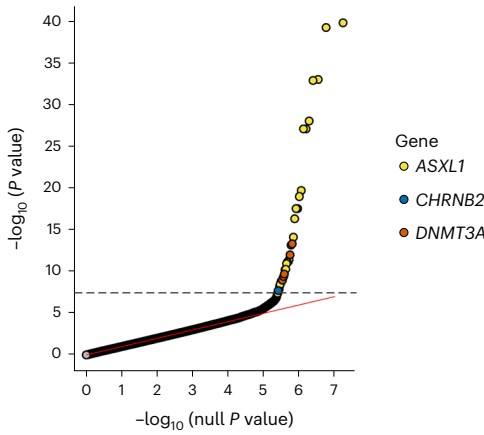

**Fig. 2 | Discovery of rare variants associated with smoking phenotypes.**
Quantile–quantile (QQ) plot of the rare variant associations (both variant and burden associations) with six smoking phenotypes (ever smoker, heavy smoker, former smoker, nicotine dependence, cig per day and age started smoking). The dashed line corresponds to the exome-wide significant threshold, $4.5 \times 10^{-8}$, determined based on a 1% FDR correction applied across all the associations ($n$ tests = 8,417,987).

six smoking phenotypes. Applying a false detection rate (FDR) of 1% (corresponding $P$ value = $4.5 \times 10^{-8}$), we identified 35 significant associations implicating three genes: *ASXL1*, *DNMT3A* and *CHRNB2* (Fig. 2, Supplementary Fig. 1 and Supplementary Table 4). Although these results were based on analyses in which individuals of all ancestries were pooled together, we found that the results were highly similar to those from a cross-ancestry meta-analysis or a meta-analysis involving only individuals of European ancestry, suggesting that the results were not influenced by population stratification (Supplementary Fig. 2).

### Associations of rare variants in *CHRNB2*
The primary phenotype that discovered the *CHRNB2* association was heavy smoker, where cases were individuals who smoked at least ten cigarettes per day either currently or formerly ($n$ = 110,494), and controls were individuals who have never smoked in their lifetime ($n$ = 374,842). The strongest association was observed for pLOF-plus-missense burden (an aggregate of pLOF and likely deleterious missense variants in *CHRNB2* with MAF < 0.001), for which the odds of being a heavy smoker were significantly lower in carriers than in non-carriers (OR = 0.65; CI = 0.56–0.76; $P$ = $1.9 \times 10^{-8}$). The rare variant burden association was independent of any nearby common variant associations with $P$ < 0.01 (Supplementary Fig. 3 and Methods), and the effect estimates were consistently in the protective direction across the three cohorts that contributed to the meta-analysis (Fig. 3). The protective association of *CHRNB2* pLOF-plus-missense burden with heavy smoking was observed irrespective of how we defined heavy smoking (Supplementary Fig. 4). Furthermore, the protective association was also seen for the ever smoker phenotype (where individuals who ever smoked regularly in their lifetime were defined as cases, $n$ = 345,805) but was less significant than for the heavy smoker phenotype, despite a relatively larger sample size, highlighting the importance of phenotype specificity in gene discovery (Extended Data Fig. 1). However, when considering pLOF-only burden (an aggregate of pLOF variants in *CHRNB2* with MAF < 0.001), which provides the strongest evidence on the direction of the association, the association reached at least a nominal level of significance ($P$ < 0.05) only for the ever smoker phenotype but not for the heavy smoker phenotype, likely because the ever smoker phenotype captured more pLOF carriers (281 carriers) than the heavy smoker phenotype (174 carriers), suggesting that a larger sample size at the expense of phenotype specificity is also valuable, particularly at the rarer end of the allele-frequency spectrum.

We next studied the association of *CHRNB2* pLOF-plus-missense burden with a range of secondary smoking phenotypes, mainly derived from UK Biobank (UKB)[27] participants' responses to a lifestyle questionnaire related to smoking (Methods). The overall association pattern was in line with our main finding that rare pLOF and likely deleterious missense variants in *CHRNB2* in aggregate confer protection against smoking addiction (Extended Data Fig. 2 and Supplementary Table 5). We also studied the burden associations with a curated list of binary and quantitative health phenotypes related to smoking and observed suggestive associations, all in the protective direction, for example, emphysema (OR = 0.45; CI = 0.28–0.71; $P$ = $6.9 \times 10^{-4}$), chronic obstructive pulmonary disease (COPD; OR = 0.80; CI = 0.62–1.03; $P$ = 0.08) and family history of lung cancer (OR = 0.84; CI = 0.69–1.01; $P$ = 0.06) (Extended Data Fig. 2).

No individual pLOF or missense variants in *CHRNB2* surpassed the study-wide significance threshold, suggesting that our sample sizes were still underpowered to capture single-variant associations. Using a leave-one-variant-out (LOVO) burden analysis[28] (Methods), we identified a missense variant (rs202079239, Arg460Gly) that contributed the most to the pLOF-plus-missense burden association in the UKB (Fig. 4a and Supplementary Table 6). Importantly, even after excluding Arg460Gly, the burden association was still nominally significant with a protective OR (OR = 0.71; CI = 0.57–0.88; $P$ = 0.001), suggesting that other variants in the burden mask also contributed to the association (Supplementary Table 6). Additionally, the Arg460Gly variant independently showed a moderately significant protective association with the heavy smoker phenotype (OR = 0.56; CI = 0.43–0.72; $P$ = $1.1 \times 10^{-5}$). We found that this variant has drifted to a higher frequency in Finns (gnomAD[29] MAF = 0.0018) compared to non-Finnish Europeans (gnomAD MAF = 0.00038; Fig. 4b). Statistical power increases with MAF; hence we expected that the protective association of Arg460Gly with smoking or related phenotypes might be detectable in FinnGen[30], a population-based cohort in Finland, despite its sample size being smaller than that of the UKB. A selective exploration of Arg460Gly with smoking, substance use and smoking-related lung disease phenotypes in the publicly available data from the FinnGen research project (freeze version 7) revealed significant enrichment for protective associations (hypergeometric test for enrichment, $P$ = 0.03; Fig. 4c,d and Supplementary Table 7). At least two phenotypes showed nominally significant ($P$ < 0.05) protective associations: substance-use disorder (excluding alcohol) (OR = 0.39; CI = 0.21–0.73; $P$ = 0.003) and COPD (OR = 0.69; CI = 0.49–0.96; $P$ = 0.03). Therefore, by exploiting the natural phenomenon of genetic drift in an isolated population[30], we were able to validate the protective association of *CHRNB2* with smoking-related phenotypes in an independent cohort.

### Associations of common variants near *CHRNB2*
Common variant associations by themselves often do not pinpoint the causal gene(s); when they do, they mostly bring limited insights into the druggability of the gene. However, when interpreted along with rare coding variant associations, they can offer valuable insights. To this end, we searched for any known common variant GWAS signals near *CHRNB2* that were reported previously for smoking-related traits. Liu et al.[16] have reported a GWAS association with cig per day near *CHRNB2* where the fine-mapped 95% credible set contained a single variant, rs2072659, located within the 3′ untranslated region (UTR) of *CHRNB2*. This variant showed significant ($P$ < 0.05) associations in our dataset with multiple smoking phenotypes including heavy smoker (OR = 0.96; CI = 0.94–0.98; $P$ = $5.3 \times 10^{-6}$), all in the protective direction (Fig. 5a). In a phenome-wide association study (PheWAS) of this variant across 7,469 phenotypes in two of the large cohorts (UKB and Geisinger Health System (GHS)), the strongest association was with smoking (Fig. 5b). In addition, seven of the top ten associations were with smoking-related phenotypes, all in the protective direction.

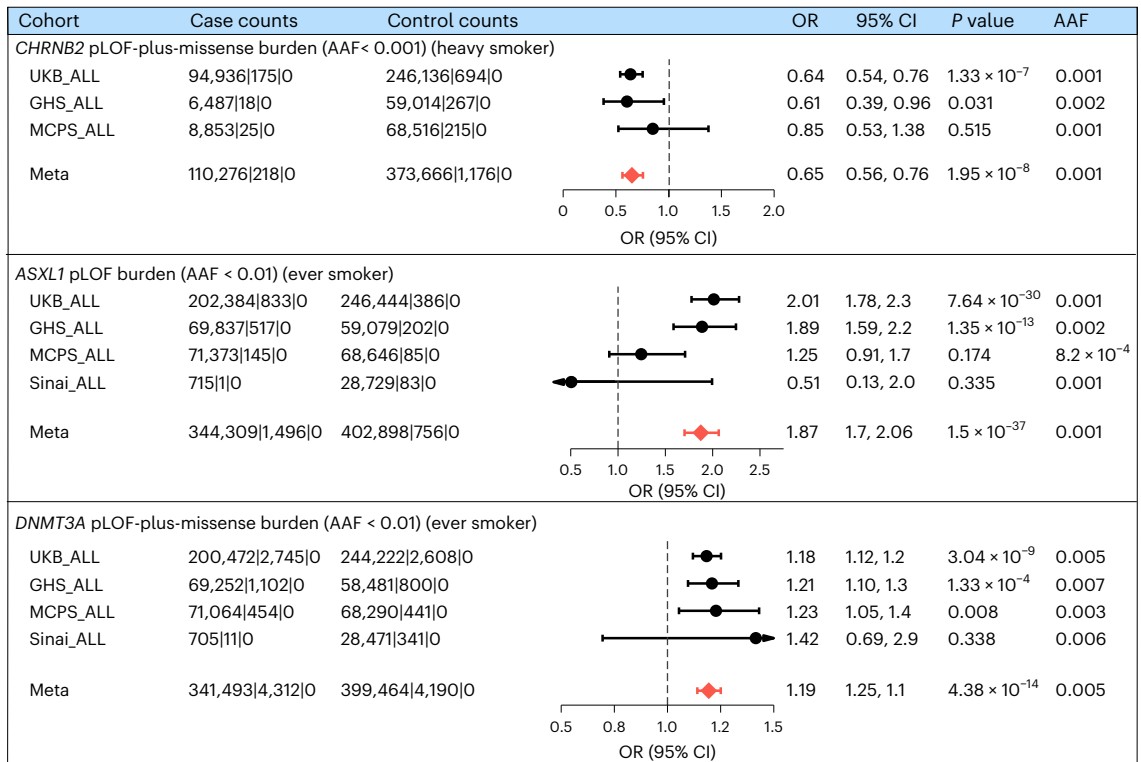

**Fig. 3 | Forest plots of the top burden–trait associations of the significant genes.** Cohort-level and meta-analysis summary statistics of the most significant burden–trait associations for each of the three exome-wide significant genes are summarized using forest plots. The ORs and 95% CIs are plotted. The columns 'case counts' and 'control counts' show the case and control sample sizes, respectively, broken down to the number of carriers of the homozygous reference, heterozygous and homozygous alternative genotypes. For burden definitions, refer to Supplementary Table 2. ALL, all ancestries; AAF, alternative allele frequency (combined frequency of all the variants aggregated in the burden mask).

## Associations of clonal hematopoiesis of indeterminate potential mutations in *ASXL1* and *DNMT3A*

Among the three exome-wide significant genes, *ASXL1* and *DNMT3A* showed the strongest associations with most of the smoking phenotypes (Figs. 2 and 3, Extended Data Figs. 3 and 4 and Supplementary Tables 4 and 5). However, both *ASXL1* and *DNMT3A* are known to accumulate somatic mutations in circulating blood cells with increasing age in the general population, a phenomenon described as clonal hematopoiesis of indeterminate potential (CHIP)[31]. When the DNA source for exome sequencing is peripheral blood, standard exome variant-calling workflows capture CHIP mutations along with germline variants[32,33]. We have previously reported a comprehensive genetic analysis of CHIP, in which we systematically called somatic variants in participants of the UKB and the GHS cohorts and studied their germline associations[33]. It is well known that smoking is strongly associated with CHIP[34,35], and the association of *ASXL1* CHIP mutations with smoking in the UKB has been previously reported[35]. Hence, we were not surprised to learn that the *ASXL1* and *DNMT3A* associations were driven by CHIP mutations, which we confirmed through burden analyses based on burden masks with and without CHIP mutations and association analyses of the variant allele fraction (VAF) of CHIP mutations with smoking phenotypes (Fig. 6, Extended Data Fig. 5 and Supplementary Note). As was previously proposed, the association of CHIP mutations with smoking phenotypes suggests that smoking offers a clonal advantage to certain CHIP mutations, although the underlying mechanisms have yet to be understood. Also, our findings echo the caution previously raised by many in relation to using exome-sequencing data based on blood samples to establish genetic diagnoses for Mendelian diseases in adults[36,37] (Supplementary Note).

## Association of rare variants at known GWAS loci

Two of the strongest genetic risk loci for smoking that were identified early in the GWAS timeline were locus 15q25.1, containing three nicotine acetylcholine receptor (nAChR) genes (*CHRNA5*, *CHRNA3* and *CHRNB4*)[38,39], and locus 19q13.2, containing a cluster of cytochrome P450 enzyme-coding genes (CYP2A, CYP2B and CYP2F subfamilies); both strongly influence the number of cigarettes smoked per day[40,41]. Although none of the genes were significant at the exome-wide level in our analysis, given their strong biological links to smoking, we explored these loci for evidence of any subthreshold rare variant associations. At the cytochrome P450 locus, we found little evidence for rare variant associations beyond the known common variant signals (Extended Data Fig. 6a and Supplementary Table 11). However, we observed nominal rare variant gene burden associations with cig per day at locus 15q25.1, implicating all three nAChRs (*CHRNA5*, *CHRNA3* and *CHRNB4*) with effect sizes larger than those observed for common variants (Extended Data Fig. 6b). Notably, the largest effect size was observed for the *CHRNB4* pLOF-only rare variant burden, where the 13 pLOF carriers smoked on average ~6.8 cigarettes per day more than non-carriers ($\beta = 0.68$ s.d.; CI = 0.17–1.18; $P = 0.008$; Extended Data Fig. 6c). This effect size is approximately three to four times larger than the largest effect sizes observed for *CHRNA5* ($\beta = 0.23$; CI = 0.05–0.40; $P = 0.01$) and *CHRNA3* ($\beta = 0.16$; CI = 0.02–0.31; $P = 0.03$) pLOF-only rare variant burden and ~7.5 times larger than that for rs16969968 (approximately one cigarette more; $\beta = 0.09$; CI = 0.09–0.10; $P = 3.8 \times 10^{-125}$), a well-characterized common risk variant at this locus (Supplementary Table 11). Power calculations based on observed effect sizes suggest that these associations will likely emerge as significant at the genome-wide level when the sample size for ExWAS of the cig per day phenotype reaches between 300,000 and 500,000 (Extended Data Fig. 7).

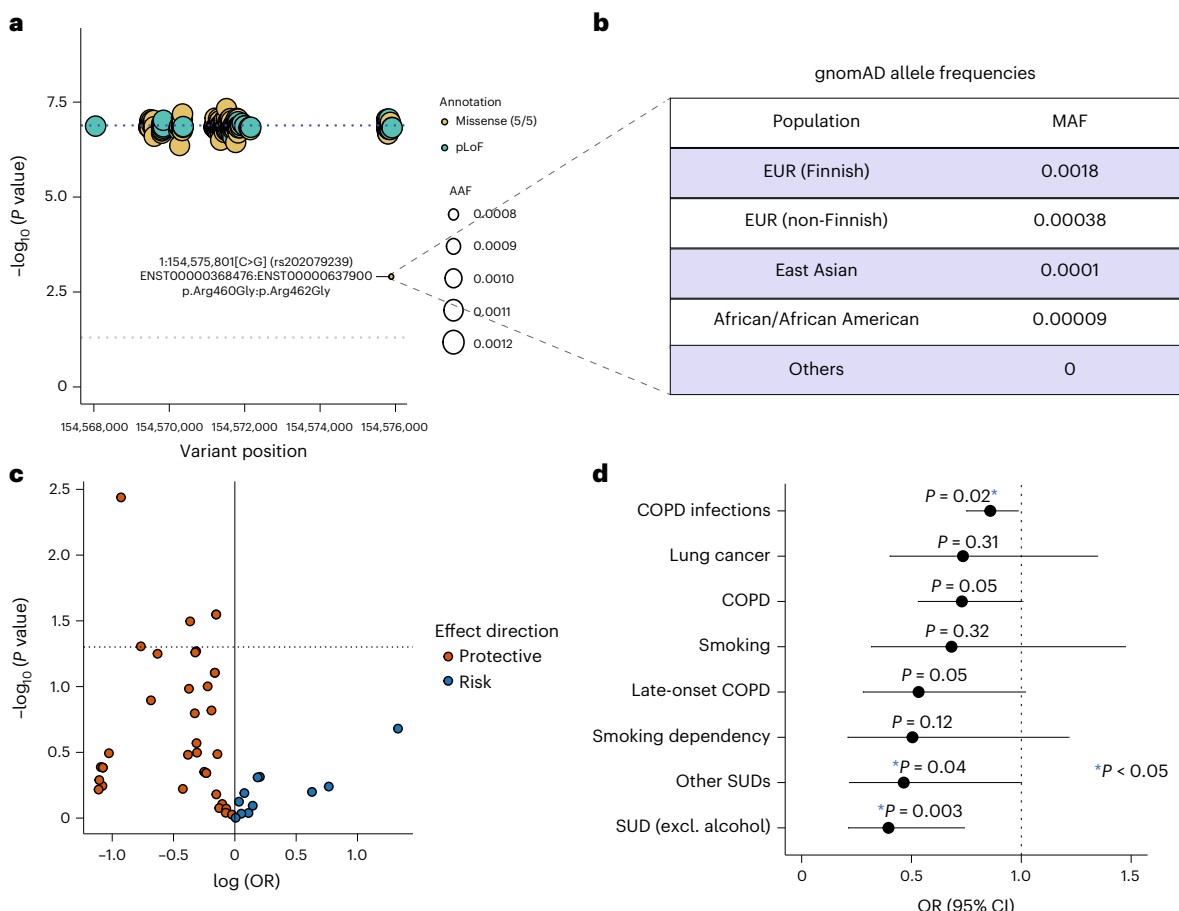

**Fig. 4 | A Finnish-enriched missense variant contributes most to the *CHRNB2* burden association. a**, Results from LOVO analysis (Methods) of the *CHRNB2* pLOF-plus-missense burden (AAF < 0.001) in the UKB. The LOVO *P* values are plotted against the variant positions. The dashed blue line corresponds to the *P* value of the full burden association. The dashed gray line corresponds to *P* = 0.05. **b**, MAFs of Arg460Gly (rs202079239) in different populations in the

gnomAD database. EUR, European ancestry. **c**, Volcano plot showing the PheWAS associations of Arg460Gly with smoking-related phenotypes in the FinnGen database. The dashed line corresponds to *P* = 0.05. **d**, ORs and 95% CIs of selected phenotype associations of Arg460Gly in the FinnGen database are displayed. Excl., excluding, SUD, substance use disorder.

Previous exome studies have shown that rare variant associations are enriched near GWAS loci for many human diseases and traits[21,42]. Hence, we analyzed the burden associations, focusing only on genes mapped to GWAS loci[19] (Methods). We observed no significant rare variant burden associations other than the association of *CHRNB2* pLOF-plus-missense burden with the heavy smoker phenotype (Extended Data Fig. 8). The results suggested that our current sample sizes are underpowered to capture the convergence between common and rare variant associations at the known smoking GWAS loci.

**Cross-ancestry and ancestry-specific GWAS**

We first performed GWAS for the six primary smoking phenotypes in individuals of European ancestries and used these results to analyze SNP-based heritability (SNP-$h^2$) and genetic correlations using a European ancestry-based linkage disequilibrium (LD) reference panel[43]. Our SNP-$h^2$ estimates were comparable to previously reported estimates[16] (Supplementary Fig. 5a and Supplementary Table 12). Also, our GWAS results showed strong genetic correlations with the previous GWAS results[16] (Supplementary Fig. 5b and Supplementary Table 13), which suggests high reproducibility of the polygenic signals of the studied smoking phenotypes. Also, we observed moderate-to-large genetic correlations across our six phenotypes, suggesting shared genetic architecture across the phenotypes (Supplementary Fig. 5c and Supplementary Table 14).

Next, we performed cross-ancestry GWAS meta-analyses for the six primary smoking phenotypes. Across all the phenotypes, in total, we identified 328 LD-independent loci, of which a majority (94%) are known. This was expected, given that a GWAS with a much larger sample size has been published before[16] (Supplementary Fig. 6a–f and Supplementary Table 16). Among the new loci, an X chromosome locus that we identified for nicotine dependence deserves special mention, as it implicates a nicotinic receptor-related gene. This locus, Xq22.1, harbors *TMEM35A* (the closest gene to the index variant), also referred to as *NACHO* (new acetylcholine receptor chaperone); this gene encodes a molecular chaperone protein that is involved in the assembly of α7, α6β2 and α6β2β3 nAChRs[44]. Mice lacking *Tmem35a* develop hyperalgesia[44], and we observed that the index variant at this locus is also associated with increased intake of oxycodone, an analgesic medication, in the UKB (OR = 1.58; *P* = 0.0001; data from https://www.opentargets.org)[45], suggesting that this locus might influence both smoking and pain phenotypes in humans.

After European ancestries, the second largest proportion (19%) of our study participants were of admixed American ancestries (AMR), mostly from the Mexico City Prospective Study (MCPS) cohort[46]. Published GWASs of smoking behavior in AMR ancestries are sparse[47]. In the AMR-specific GWAS, we identified 25 independent loci across the six phenotypes, of which 15 are known and 10 are new (Supplementary Table 16). The known loci include some of the strongest

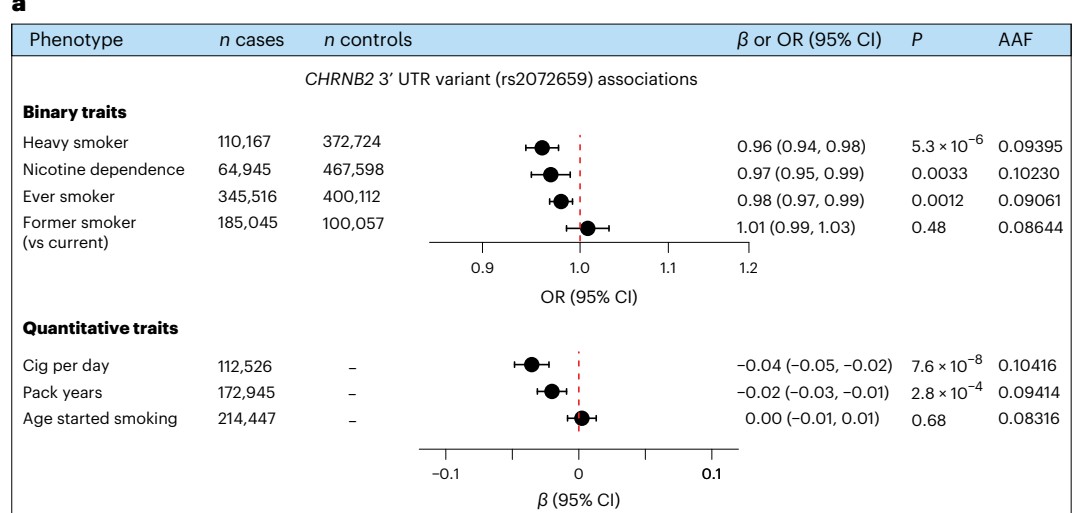

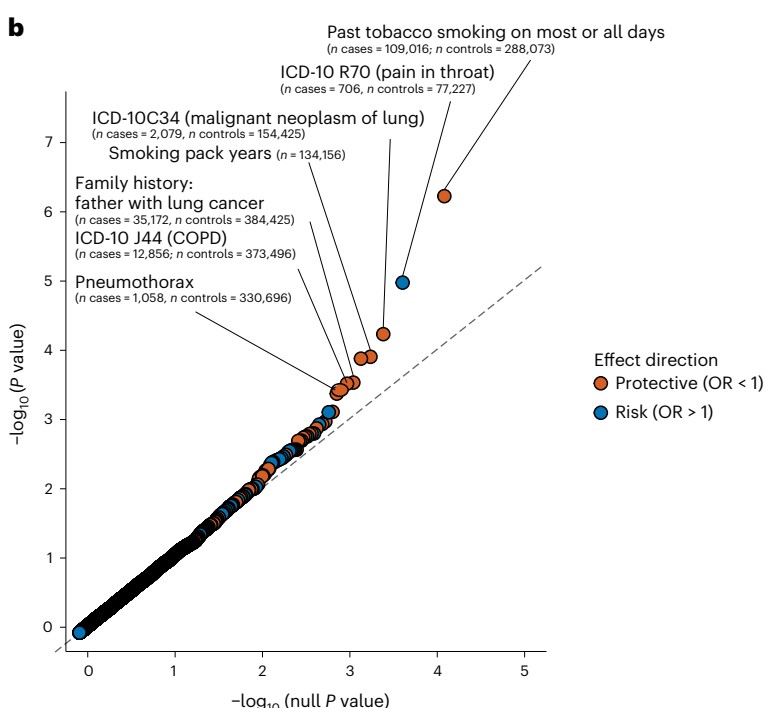

**Fig. 5 | Association of a common 3′ UTR variant with smoking. a**, Forest plots of associations of the *CHRNB2* 3′ UTR variant (rs2072659) with the major smoking phenotypes based on cross-ancestry meta-analyses (Methods); either ORs (if binary traits) or *β* estimates (in s.d. units) and their 95% CIs are plotted. **b**, QQ plot of the PheWAS associations of rs2072659 in the UKB and the GHS cohorts.

GWAS loci identified in European-specific GWAS: *CHRNA5* (ref. 39), *CHRNA4* (ref. 48), *DBH*[41], *CYP2A6* (refs. 40,41) and *NCAM1* (ref. 49) (Supplementary Table 16). In AMR ancestries, we also identified an X chromosome locus that has been previously linked to smoking in those of European ancestries[18]. Notably, at this locus (with *GPR101* in the vicinity), we identified a genome-wide significant association with the heavy smoker phenotype in the AMR-specific GWAS (rs1190734; $OR_{AMR} = 0.83$ (0.79–0.88); $P_{AMR} = 1.2 \times 10^{-11}$) but only a nominal association with the heavy smoker phenotype in the European-specific GWAS ($OR_{EUR} = 0.98$ (0.97–0.99); $P_{EUR} = 0.001$). However, the same variant showed genome-wide significant association with the cig per day phenotype in European-specific GWAS ($β_{EUR} = -0.02$; $P_{EUR} = 7.6 \times 10^{-16}$), corroborating the GWAS signal at this locus reported previously for the cig per day phenotype[18]. Whether this locus is associated with the cig per day phenotype in AMR ancestry with a larger effect size

than that in European ancestry is not clear, as we did not have this phenotype in the MCPS cohort at the time of this analysis. Nevertheless, the findings overall suggest that the *GPR101* locus influences smoking behavior in both European and AMR ancestries. Regarding the ten new loci identified in the AMR ancestries, as expected, many (seven loci) harbored variants that are relatively more common in AMR ancestries than in European ancestries, thereby offering higher statistical power for discovery; for example, at 10q21.1, an intergenic locus, we identified a genome-wide significant association with the heavy smoker phenotype where the index variant is observed in ~10% of admixed Americans but only in ~0.05% of Europeans; at 8p22 (closest gene, *C8orf48*), we identified a genome-wide significant association with the ever smoker phenotype, where the index variant is observed in ~30% of admixed Americans but only in ~7% of Europeans (Supplementary Table 16).

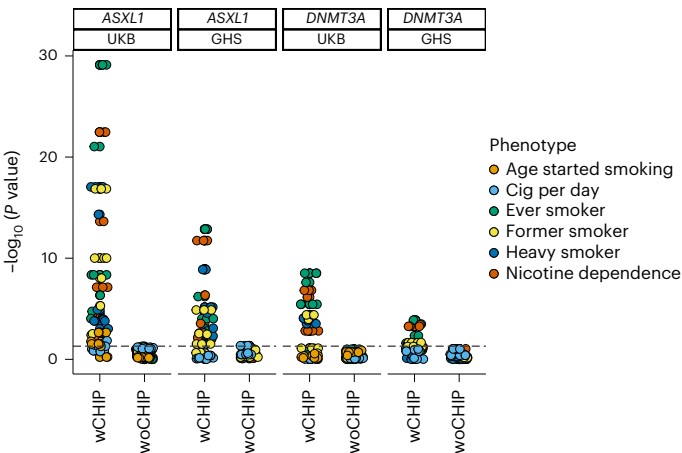

**Fig. 6 | Association of *ASXL1* and *DNMT3A* CHIP mutations with smoking.**
We constructed pLOF-only and pLOF-plus-missense burden masks at five
allele frequency thresholds using all variants (wCHIP) and excluding CHIP
variants (woCHIP) in the UKB and the GHS cohorts and tested their associations
with the six major smoking phenotypes using REGENIE (Methods). The burden
association *P* values are plotted, and the summary statistics including
sample sizes are provided in Supplementary Table 8. The dashed line
corresponds to the significance threshold after adjusting for multiple testing
(1% FDR correction).

## Interplay between common and rare variants

Large-scale sequencing projects provide increased power to detect
additive effects between common and rare variants for many dis-
eases and traits. For example, we have previously demonstrated an
additive effect between *GPR75* obesity-protective rare variants and
polygenic score (PGS) for obesity based on common variants[10]. We
performed a similar analysis to test whether an additive effect is also
evident for *CHRNB2* rare variants and smoking PGS. We calculated
smoking PGS for UKB participants of European ancestries based on
a GWAS of the ever smoker phenotype performed in an independ-
ent sample (a meta-analysis of GWAS and Sequencing Consortium of
Alcohol and Nicotine use (GSCAN) GWAS[19] results excluding 23andMe
and the UKB with the GWAS results of the GHS[50], one of our largest
European cohorts). First, we studied the associations of *CHRNB2*
pLOF-plus-missense burden and smoking PGS with heavy smoking
within a single regression model that included an interaction term
between the burden mask and the PGS (Methods). Both burden mask
(OR = 0.66; 95% CI = 0.56–0.79; *P* = 3.4 × 10$^{-6}$) and the PGS ($\beta$ = 0.33;
standard error (SE) = 0.004; *P* = 1 × 10$^{-300}$) were associated with heavy
smoking without a statistically significant interaction (*P* = 0.71). The
results suggest that rare variants and the PGS influence the risk of
heavy smoking independently. Second, to demonstrate the additive
effect, we binned UKB individuals into quintiles based on their smok-
ing PGS and quantified the prevalence of heavy smokers in *CHRNB2*
pLOF-plus-missense burden mask carriers (the burden mask that
showed the strongest association with the heavy smoker phenotype)
and non-carriers. The prevalence of heavy smokers increased in both
carriers and non-carriers from lower to higher PGS quintiles (Fig. 7 and
Supplementary Table 17). Importantly, within each of the quintiles, the
prevalence of heavy smokers was lower in *CHRNB2* rare variant carri-
ers than in non-carriers, demonstrating an additive effect between
PGS and rare variants. The additivity implies that the smoking PGS
modifies the penetrance of *CHRNB2* rare variants and vice versa, that
is, the protective effect of *CHRNB2* rare variant burden is attenuated
in individuals with higher PGS compared to in individuals with lower
PGS, and the risk effect of increased PGS is attenuated in rare variant
carriers compared to in non-carriers.

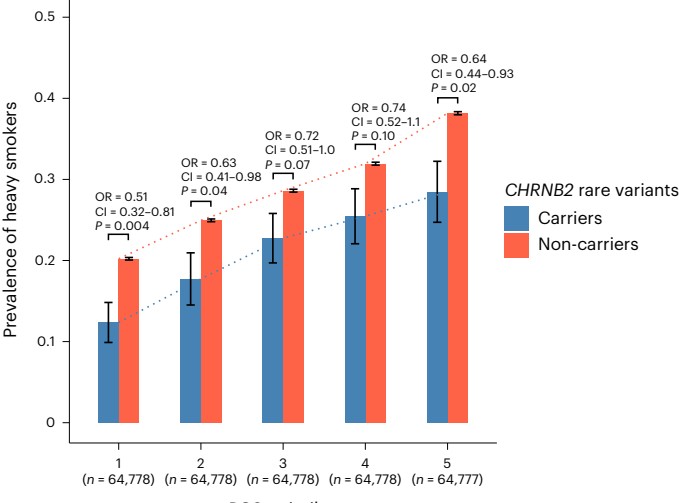

**Fig. 7 | Additive effects between *CHRNB2* rare variants and smoking PGS.**
Prevalence estimates of heavy smokers among *CHRNB2* rare variant carriers and
non-carriers within each of the five PGS quintiles in the UKB are plotted. Standard
errors of the prevalence estimates, displayed as error bars, were calculated using
the formula $\sqrt{(pq/n)}$, where *n* is the number of individuals in each group, *p* is the
prevalence of heavy smoker in the group and *q* is 1 − *p*. The PGS was based on a
GWAS meta-analysis of the ever smoker phenotype (Methods). *CHRNB2* rare
variants are those that were aggregated into the *CHRNB2* pLOF-plus-missense
(AAF < 0.001) burden mask. Statistical differences in the prevalence between
carriers and non-carriers were tested using a logistic regression analysis within
each quintile; ORs, 95% CIs and *P* values are shown.

## Discussion

GWASs of smoking behavior[16–19] based on common variants have
made tremendous progress in the field, with the recent GWAS involv-
ing more than 3 million individuals[19]. Such studies have substantially
improved our understanding of the polygenic architecture of smok-
ing phenotypes and have highlighted genes and pathways including
nAChRs, genes involved in nicotine metabolism and dopaminergic
and glutamatergic signaling[16]. However, to date, few studies based
on whole-exome- or whole-genome-sequencing data have been
reported[15,20], and they involved sample sizes insufficient to capture
associations at variant- and gene-level resolutions. Hence, our under-
standing of the contributions of rare variants to smoking behavior has
been minimal thus far. In the present study, we performed a large-scale
rare variant analysis in sample sizes that had enough power to identify
associations of a rare variant or an aggregate of rare variants with an OR
of 2.5 and above (or 0.4 and below) when there are at least 100 carriers
(Extended Data Fig. 9). The fact that our analysis revealed only one
germline association indicates that there are no 'low-hanging fruits'
for smoking in the rare variant space other than *CHRNB2*. However, we
acknowledge that this interpretation applies only to European popula-
tions, and we cannot exclude the possibility that rare variants exist that
are more frequent in other ancestries and might be discovered in the
future in similar or even smaller sample sizes than ours. Nevertheless,
we note that 25% of our samples represent non-European ancestries,
with the largest proportion (19%) representing admixed Americans[46].
However, the sample sizes, when broken down into individual ancestry
groups, are still smaller than what would be necessary to make rare
variant discoveries.

The major finding from our analysis is that individuals with
rare pLOF and likely deleterious missense variants in *CHRNB2* are at
decreased odds of smoking heavily. Although the top association
was observed for the gene burden that combined both pLOF and mis-
sense variants, the concordant protective effect sizes observed for

the pLOF-only burden strengthened our interpretation that what we observe is a loss-of-function association. This knowledge is crucial as it informs therapeutic hypotheses for drug design. Moreover, we identified a single deleterious missense variant that drifted to a higher frequency in the Finnish population, which gave us an opportunity to validate the protective associations in the FinnGen study[30]. The finding highlights the value of isolated populations to inform drug target discovery[51].

Another important finding is the convergence of rare and common variant findings of *CHRNB2*. We highlight a common 3′ UTR variant, reported in previous GWASs[16,19], that shows protective associations with multiple smoking phenotypes, suggesting that this variant likely decreases *CHRNB2* expression. Importantly, the OR of the common variant association with the heavy smoker phenotype was 0.96 as opposed to 0.65 for the pLOF-plus-missense rare variant burden. The pattern suggests a dose–response relationship between the gene and the phenotype in which varying levels of gene perturbations result in proportional effects on the phenotype. We particularly highlight the fact that this variant, although discovered in the earlier GWAS[16], did not receive attention, as it was buried underneath the hundreds of GWAS associations, reflecting an important limitation of interpreting common variant findings. However, when interpreted in the light of rare variant findings, the common variant association stood out as highly valuable, exemplifying the combined value of GWAS and ExWAS in drug target discovery. Such observations will become frequent in the future with the rapidly growing population-scale ExWAS of human diseases and traits[52].

*CHRNB2* codes for the β2 subunit of the α4β2 nAChR, which is the predominant nicotinic receptor expressed in the human brain[53]. The role of α4β2 nAChR in mediating nicotine effects has been well characterized by decades of animal studies[54,55], thanks to the pioneering work of Picciotto and colleagues who demonstrated in 1995 that deletion of the gene encoding β2 in mice abolished nicotine-mediated effects on avoidance learning and reinforcement behavior[56,57]. However, we describe human genetic evidence supporting the hypothesis that loss of *CHRNB2* protects against nicotine addiction. Importantly, the protein encoded by *CHRNB2* can be viewed as a known drug target as it is a component of the α4β2 nAChR, which, being the major nicotine receptor in the brain, has been the target of most nAChR partial agonists and antagonists developed thus far, including cytisine (an α4β2 partial agonist[58]) and varenicline (an α4β2 partial agonist and antagonist[3]). Varenicline is the current drug of choice to aid smoking cessation and was developed in 1997 by Pfizer based on the molecular structure of cytisine[2,3]. In addition to α4β2, varenicline binds to various other nAChRs in the brain including α7, α3β4 and α6β2 (ref. 59). Given the established role of α4β2 in mediating rewarding and reinforcement actions of nicotine, it is believed that the α4β2-antagonistic action of varenicline helps with smoking cessation[3]. Our finding aligns with this hypothesis, emphasizing that human genetics is useful not only to discover new drugs but also to better understand the mechanism of action of old drugs that have been in use for decades, and such knowledge can pave the way for better drug designs with greater efficacy and limited adverse effects.

Limitations of our study include small sample sizes for finer quantitative phenotypes such as cig per day, which have limited our power to capture rare variant associations of genes mediating aversive effects of nicotine (for example, *CHRNA5*) and those related to nicotine metabolism (for example, *CYP2A6*)[39,40]. As is often the case, individuals of non-European ancestries were under-represented in our study cohorts, which has limited the generalizability of the findings to all ancestries[60,61]. However, we involved a substantial number of individuals of AMR ancestries, who belong to one of the most under-represented populations in human genetic studies, a step in the right direction[46]. With growing awareness of the importance of diversity in human genetic studies, the representation of non-European ancestries is expected

to improve in future studies[60,61]. Finally, we have focused only on the coding regions of the genome captured via whole-exome sequencing, and therefore we may have missed rare variants with large effects on smoking behavior residing in noncoding regulatory regions. With the recent increase in large-scale whole-genome-sequencing efforts, rare large-effect regulatory variants influencing human diseases and traits are being discovered, and such discoveries may have the potential to lead to drug targets[62]. However, the question of whether whole-genome sequencing is a more cost-effective investment than whole-exome sequencing for drug target discovery has yet to be answered.

To conclude, we have performed a large-scale ExWAS of smoking behavior and identified a protective association between rare coding variants in *CHRNB2* and smoking. The results align with the findings from published knockout animal models and the mechanism of action of varenicline that is currently in use to aid smoking cessation and will support future therapeutic developments to treat smoking addiction.

## Online content

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

**GHS-REGN DiscovEHR collaboration**

Lance J. Adams[7], Jackie Blank[7], Dale Bodian[7], Derek Boris[7], Adam Buchanan[7], David J. Carey[7], Ryan D. Colonie[7], F. Daniel Davis[7], Dustin N. Hartzel[7], Melissa Kelly[7], H. Lester Kirchner[7], Joseph B. Leader[7], David H. Ledbetter[7], J. Neil Manus[7], Christa L. Martin[7], Raghu P. Metpally[7], Michelle Meyer[7], Tooraj Mirshahi[7], Matthew Oetjens[7], Thomas Nate Person[7], Christopher Still[7], Natasha Strande[7], Amy Sturm[7], Jen Wagner[7] & Marc Williams[7]

[7]Geisinger, Danville, PA, USA.

**Regeneron Genetics Center**

**RGC Management & Leadership Team**

Gonçalo R. Abecasis[1], Aris Baras[1], Aris Economides[1], Michael Cantor[1], Giovanni Coppola[1], Andrew Deubler[1], Aris Economides[1], Katia Karalis[1], Luca A. Lotta[1], John D. Overton[1], Jeffrey G. Reid[1], Katherine Siminovitch[1], Lyndon J. Mitnaul[1], Alan Shuldiner[1] & Adolfo Ferrando[1]

**Sequencing & Lab Operations**

Christina Beechert[1], Caitlin Forsythe[1], Erin D. Brian[1], Zhenhua Gu[1], Michael Lattari[1], Alexander Lopez[1], John D. Overton[1], Maria Sotiropoulos[1], Manasi Pradhan[1], Kia Manoochehri[1], Ricardo Schiavo[1], Raymond Reynoso[1], Kristy Guevara[1], Laura M. Cremona[1], Chenggu Wang[1], Hang Du[1] & Sarah E. Wolf[1]

**Clinical Informatics**

Amelia Averitt[1], Nilanjana Banerjee[1], Michael Cantor[1], Dadong Li[1], Sameer Malhotra[1], Deepika Sharma[1], Justin Mower[1], Jay Sundaram[1], Aaron Zhang[1], Sean Yu[1], Mudasar Sarwar[1] & Jeffrey C. Staples[1]

**Genome Informatics & Data Engineering**

Xiaodong Bai[1], Lance Zhang[1], Sean O'Keeffe[1], Andrew Bunyea[1], Lukas Habegger[1], Suganthi Balasubramanian[1], Suying Bao[1], Boris Boutkov[1], Gisu Eom[1], Lukas Habegger[1], Alicia Hawes[1], Olga Krasheninina[1], Rouel Lanche[1], Adam J. Mansfield[1], Evan Edelstein[1], Sujit Gokhale[1], Alexander Gorovits[1], Evan K. Maxwell[1], Ju Guan[1], George Mitra[1], Janice Clauer[1], Mona Nafde[1], Vrushali Mahajan[1], Razvan Panea[1], Koteswararao Makkena[1], Krishna PawanPunuru[1], Benjamin Sultan[1], Sanjay Sreeram[1], Tommy Polanco[1], Ayesha Rasool[1], Jeffrey G. Reid[1], William J. Salerno[1] & Kathie Sun[1]

**Analytical Genetics and Data Science**

Joshua Backman[1], Anthony Marcketta[1], Bin Ye[1], Lauren Gurski[1], Nan Lin[1], Gonçalo R. Abecasis[1], Jonathan Marchini[1], Jan Revez[1], Yuxin Zou[1], Jack Kosmicki[1], Jonathan Ross[1], Joelle Mbatchou[1], Andrey Ziyatdinov[1], Kyoko Watanabe[1], Eli Stahl[1], Akropravo Ghosh[1], Lei Chen[1], Rujin Wang[1], Adam Locke[1], Carlo Sidore[1], Arden Moscati[1], Lee Dobbyn[1], Eric Jorgenson[1], Blair Zhang[1], Christopher Gillies[1], Michael Kessler[1], Maria Suciu[1], Timothy Thornton[1], Priyanka Nakka[1], Sheila Gaynor[1], Tyler Joseph[1], Benjamin Geraghty[1], Anita Pandit[1], Joseph Herman[1], Sam Choi[1], Peter VandeHaar[1], Liron Ganel[1], Kuan-Han Wu[1], Aditeya Pandey[1], Kathy Burch[1], Adrian Campos[1], Scott Vrieze[1], Sailaja Vedantam[1], Charles Paulding[1] & Amy Damask[1]

**Therapeutic Area Genetics**

Ariane Ayer[1], Aysegul Guvenek[1], George Hindy[1], Giovanni Coppola[1], Jan Freudenberg[1], Jonas Bovijn[1], Katherine Siminovitch[1], Luca A. Lotta[1], Manav Kapoor[1], Mary Haas[1], Moeen Riaz[1], Niek Verweij[1], Olukayode Sosina[1],

 **1147**

# Article

Parsa Akbari[1], Priyanka Nakka[1], Sahar Gelfman[1], Sujit Gokhale[1], Tanima De[1], Veera M. Rajagopal[1], Alan Shuldiner[1], Bin Ye[1], Gannie Tzoneva[1], Jin He[1], Adolfo Ferrando[1], Silvia Alvarez[1], Kayode Sosina[1], Neelroop Parikshak[1], Jacqueline Otto[1], Anna Alkelai[1], Vijay Kumar[1], Peter Dombos[1], Amit Joshi[1], Sarah Graham[1], Luanluan Sun[1], Antoine Baldassari[1], Jessie Brown[1], Cristen J. Willer[1], Arthur Gilly[1], Hossein Khiabanian[1], Brian Hobbs[1], Billy Palmer[1] & Juan Rodriguez-Flores[1]

**Research Program Management & Strategic Initiatives**

Esteban Chen[1], Jaimee Hernandez[1], Marcus B. Jones[1], Michelle G. LeBlanc[1], Jason Mighty[1], Nirupama Nishtala[1], Nadia Rana[1] & Jennifer Rico-Varela[1]

**Strategic Partnerships & Business Operations**

Randi Schwartz[1], Thomas Coleman[1], Alison Fenney[1], Jody Hankins[1], Ruan Cox[1] & Samuel Hart[1]

## Methods

### Participating cohorts

**UK Biobank.** The UKB is an open-access, large population cohort of 500,000 individuals established in the United Kingdom[27,63]. The participants were, in general, community-dwelling middle-aged to old-aged volunteers who were recruited between 2006 and 2010 through invitations sent by mail[63]. The age of the participants ranged between 40 and 69 years at the time of recruitment. A deep set of phenotypes has been collected from the participants prospectively, including physical, biochemical and multimodal imaging measures, disease history based on electronic health records (EHRs) and a wide range of environmental measures obtained via touchscreen and web-based questionnaires. The smoking phenotypes that we studied in this project were based on the information collected through lifestyle and environment touchscreen questionnaires (data field category 100058). The health-related phenotypes that we studied including the history of lung and vascular diseases are based on ICD-10 codes from the EHRs or self-reported or a combination of both.

**Geisinger Health System.** The GHS participants come from Geisinger's MyCode Community Health Initiative, which was established in 2007 to create a biorepository for research projects investigating the molecular and genetic bases of health and disease[50,64]. The participants were patients enrolled in the health care system who consented to participate in the MyCode initiative and gave access to their EHRs. The smoking phenotypes that we studied were based on the clinical history of smoking available in the EHR. Finer details on the smoking behavior such as the number of cigarettes smoked per day, age started smoking, etc. were available for a subset of patients through spirometry questionnaires available in the EHR.

**Mexico City Prospective Study.** The MCPS is a large prospective cohort of 150,000 individuals recruited between 1998 and 2004 with a major aim to investigate the known and new risk factors for mortality in individuals of Mexican descent[46,65]. The participants were residents of the Coyoacan and Iztapalapa districts of Mexico City. Phenotype data including information on smoking behavior were collected through house-to-house visits through interviewer-administered questionnaires.

**Sinai.** The Sinai participants were from the BioMe Biobank Program of the Charles Bronfman Institute for Personalized Medicine at the Mount Sinai Medical Center established in 2007 (ref. 66). The BioMe participants are patients enrolled in the Mount Sinai health system who consented to participate in the BioMe initiative and gave access to their EHRs. The smoking phenotypes that we studied were derived from the EHR.

### Ethical approval and informed consent

All study participants have provided informed consent, and all participating cohorts have received ethical approval from their respective institutional review board. The UKB project has received ethical approval from the Northwest Centre for Research Ethics Committee (11/NW/0382)[21,27]. The work described here has been approved by the UKB (application no. 26041)[21]. The GHS project has received ethical approval from the Geisinger Health System Institutional Review Board under project no. 2006-0258 (refs. 50,64). The MCPS has received ethical approval from the Mexican Ministry of Health, the Mexican National Council for Science and Technology, the UNAM and the University of Oxford[46,65]. The BioMe biobank has received ethical approval from the institutional review board at the Icahn School of Medicine at Mount Sinai[66].

### Phenotype definitions

We defined six phenotypes for the primary analysis: (1) ever smoker: cases were those who ever smoked regularly (including both former

and current smokers), and controls were those who never smoked in their lifetime; (2) heavy smoker: cases were those who smoked ten or more cigarettes per day (including both former and current smokers), and controls were those who never smoked in their lifetime; (3) former smoker: cases were those who smoked in the past but not at the present, and controls were current smokers; (4) nicotine dependence: cases were those who had an ICD-10 F17 diagnosis in the EHR, and controls were those who did not have an ICD-10 F17 diagnosis; (5) cig per day: number of cigarettes smoked per day in both current and former smokers; (6) age started smoking: age when the person first started smoking.

In addition to the six primary phenotypes, we also studied a set of secondary smoking phenotypes primarily derived from the smoking lifestyle questionnaire data in the UKB (data field category 100058). We also studied a selected list of disease phenotypes related to smoking, namely lung cancer (ICD-10 C34), COPD (ICD-10 J44), emphysema (ICD-10 J43), chronic bronchitis (ICD-10 J42), peripheral arterial disease (ICD-10 I73), coronary artery disease (ICD-10 I25) and myocardial infarction (ICD-10 I21).

### Exome sequencing and variant calling

The exomes of individuals from all participating cohorts were sequenced at the RGC. Exome-sequencing and variant-calling workflows followed for each of the participating cohorts are described in detail elsewhere[10,21,46,64,67]. Briefly, the DNA source for exome sequencing in all the cohorts was peripheral blood. The DNA samples were first enzymatically fragmented into 200-bp DNA libraries, to which 10-bp barcodes were added to facilitate multiplexed operations. Exome regions containing DNA fragments were captured overnight using a modified version of the xGen probe from Integrated DNA Technologies. The captured fragments were then amplified by PCR and sequenced in a multiplexed manner using 75-bp paired-end reads on the Illumina NovaSeq 6000 platform. On average, 20× coverage was achieved for more than 90% of the target sequences in 99% of the samples.

Sequenced reads were mapped to the hg38 reference genome using BWA-MEM to create BAM files. Duplicated reads were marked for exclusion using the Picard tool. Next, variant calling was performed at individual sample levels using the WeCall variant caller to create per-sample gVCF files to enable a sample-level filter. Data from samples with low sequence coverage (<85% of the targeted bases achieving >20× coverage), excess heterozygosity, disagreement between genetic and reported sex, disagreement between exome and array genotype calls and genetic duplicates were removed. The remaining high-quality gVCF files were merged into a single project-level VCF (pVCF) file using the GLnexus joint genotyping tool. A further variant-level filter was applied to the multi-sample pVCF file. SNVs with read depth <7 and indels with read depth <10 were removed. Also, variants without either at least a single homozygous genotype or a single heterozygous genotype with allele balance ratio ≥0.15 (≥0.20 if indel) were removed. The quality-controlled pVCF files were then converted to analysis-ready PGEN format using PLINK version 2.

### Variant annotation

Variants called from exome-sequencing data were annotated using the SnpEff tool[68]. Each variant was assigned the most severe consequence across all the protein-coding transcripts for which start and end positions were defined according to Ensembl release 85. Variants with any of the following annotations: stop gain, start lost, splice donor, splice acceptor, stop lost and frameshift corresponding to the non-ancestral allele were annotated as pLOF variants. Missense deleteriousness was predicted using five different algorithms, namely SIFT[69], PolyPhen-2 HDIV and PolyPhen-2 HVAR[70], LRT[71] and MutationTaster[72], and missense variants that were predicted to be deleterious by all five algorithms were annotated as 'likely deleterious' variants.

## Genotyping and imputation

Genotyping was performed using DNA genotyping arrays that varied from cohort to cohort and are reported in detail in cohort-specific publications[27,46,64]. Briefly, UKB participants were genotyped using the Applied Biosystems UK BiLEVE Axiom Array or the Applied Biosystems UKB Axiom Array; GHS participants were genotyped using either the Illumina Infinium OmniExpressExome or the Global Screening Array; and MCPS and Sinai participants were genotyped using the Global Screening Array. Standard quality-control procedures were followed to retain only high-quality genotyped variants, which were then used for imputing common variants using the TOPMed LD reference panel[73]. For all cohorts, imputation was performed in the TOPMed Imputation Server by uploading the quality-controlled genotypes in randomized batches. Following imputation, we retained only variants with MAF > 0.01 and imputation INFO score > 0.8 for the analysis reported in the current study. After all quality control, the final number of common variants included in the cross-ancestry meta-analyses ranged from ~6.7 million for the ever smoker phenotype to ~14 million variants for the cig-per-day phenotype (the final number of variants decreased as expected with increases in the number of cohorts included in the meta-analyses). Appropriate variables for the genotyping arrays and the imputation batches were used as covariates in all analyses of imputed variants.

## Genetic ancestry inference

Genetic ancestries of the individuals from all participating cohorts were quantified using a set of common variants that were genotyped directly using the genotyping arrays[21]. We first computed principal components (PCs) in HapMap3 individuals using the publicly available genotype reference panel[74]; only high-confidence variants (MAF > 0.10, genotype missingness < 5% and Hardy–Weinberg equilibrium test $P > 1 \times 10^{-5}$) that were common between our dataset and HapMap3 were used for PC calculations. PCs were first computed in the HapMap3 samples on which the rest of the samples were projected. Individuals were assigned to one of five ancestral groups, namely, Europeans, Africans, AMR, East Asians and South Asians, if their likelihood of belonging to a particular ancestry was >0.3; the likelihood estimate was calculated using a kernel density estimator trained on the HapMap3 PCs[21].

## Genetic association analysis

Genetic association analyses were performed within each of the cohorts separately using REGENIE software[28], and the results were then meta-analyzed together using an inverse-variance-weighted approach using METAL software[75]. REGENIE uses a two-step whole-genome regression framework that controls for population stratification and sample relatedness in a cost-effective and computationally efficient manner. Briefly, in step 1, REGENIE computes trait-prediction values (also called local PGS) using a sparse set of genotypes, which are typically the array genotypes. In step 2, REGENIE computes the variant associations with phenotypes using either logistic or linear regression, where the trait-prediction values computed in step 1 are included as covariates along with other covariates, namely the first 20 genetic PCs computed using common variants, the first 20 genetic PCs computed using rare variants, age, age squared, sex, an interaction term between age and sex and genotyping batches. Specifically, for binary traits with imbalanced case–control ratios, REGENIE uses a fast Firth regression, which has been shown to perform better than saddlepoint-approximation correction used in the logistic mixed-model approach implemented in software such as SAIGE[76]. For burden analysis, REGENIE first creates a pseudo-genotype, described as a burden mask, by collapsing a set of variants (see Supplementary Table 2 for the different burden definitions used) into a single categorical variable and then treats this burden mask in the same manner as a variant genotype to compute association statistics.

For the top burden associations, we performed a sensitivity analysis called LOVO implemented in REGENIE. To perform LOVO, REGENIE creates a series of burden masks iteratively for a given set of variants,

where, during each iteration, one variant is left out of the burden mask. The created burden masks are then tested for association with the phenotype of interest. Variants that contribute substantially to the burden association will cause a large drop in the statistical significance when left out. Therefore, such an approach can isolate variants that are mainly driving the association and can help evaluate whether a burden association is driven by multiple variants or only a single variant; this is important, as, in the latter, the inferred effect direction cannot be attributed to all variants that were included in the burden mask.

For the top burden associations, we also tested whether the associations were driven by any nearby common variant signals. For this, we iteratively included the most significant common variant observed within 1 Mb on either side of the gene start as a covariate in the REGENIE regression analysis until no nearby common variants with $P < 0.01$ were observed. The burden results from the conditional analysis in each of the cohorts were then meta-analyzed together.

## FinnGen analysis

We downloaded the associations of variant rs202079239 with 3,095 disease endpoints in the FinnGen database using their web browser (https://r7.finngen.fi/variant/1-154575801-C-G)[30]. Through a string search, we extracted associations related to smoking, substance abuse, addiction, COPD and other lung diseases. To test for enrichment of protective associations (OR < 1) in the extracted phenotypes, we did a hypergeometric test using the 'phyper' function implemented in the R base package by passing the following values: $q = 36$ (number of associations with OR < 1 among the smoking-related phenotypes), $m = 2{,}018$ (number of associations with OR < 1 among all phenotypes), $n = 1{,}077$ (number of associations with OR > 1 among all phenotypes) and $k = 47$ (total number of smoking-related phenotypes extracted).

## Association of rare variant burden at known GWAS loci

The most recent GWAS by Saunders et al. has identified 1,647 loci associated with one or more smoking traits, and, furthermore, the authors have mapped a set of 'high-priority genes' through statistical fine mapping[19]. Leveraging these results, we analyzed rare variant burden associations with our six primary smoking phenotypes focused on two gene sets: high-priority genes ($n$ genes = 788) and a broader list of genes that are located close to any of the 1,647 GWAS loci reported by Saunders et al. ($n$ genes = 1,177)[19]. Similar to our primary analysis, we studied pLOF-only and pLOF-plus-deleterious missense variant burden at five allele frequency thresholds for each of the genes. We applied an FDR of 1% to correct the $P$ values for multiple testing.

## CHIP mutation analysis

We identified CHIP mutations in the exome-sequencing data of UKB and GHS participants using a somatic mutation-calling pipeline, which we have described in detail in a previous publication focused on CHIP[33]. Briefly, we used the somatic mutation caller Mutect2, which uses variant mapping and allele-frequency measures to call somatic mutations against a background of germline variants and sequencing errors. CHIP mutation calls were then refined using exome data of a set of reference individuals without somatic mutations (sampled from the lower tail of the age distribution). This was followed by a series of quality-control filtering to identify a final set of highly confident CHIP mutations. In the current work, we studied only the CHIP mutations identified in the eight most recurrent CHIP genes (*DNMT3A*, *TET2*, *ASXL1*, *PPM1D*, *TP53*, *JAK2*, *SRSF2* and *SF3B1*)[33].

To test whether the ExWAS associations of *ASXL1* and *DNMT3A* are driven by CHIP mutations, we constructed gene burden masks that excluded CHIP mutations and performed burden association tests using REGENIE and compared with the results based on burden masks that included all rare variants. Furthermore, we constructed burden masks for all eight recurrent CHIP genes using only the CHIP mutations and performed burden analysis using REGENIE. We also tested

the associations of VAF of the CHIP mutations with the six smoking phenotypes in a merged genetic dataset of CHIP mutation carriers in the UKB ($n$ = 28,348) and the GHS ($n$ = 11,063) cohorts. We aggregated the VAF estimates for CHIP mutations within each (and across all) of the eight genes and tested their associations with smoking phenotypes through regression analysis adjusted for age, sex, the first ten genetic PCs and a dummy variable for the cohort of origin.

## Identification of independent known and new GWAS loci

To define approximate LD-independent GWAS signals, we used conditional and joint analysis (COJO) implemented in the GCTA software[77]. For the LD reference, we used individual-level genotype data of 10,000 randomly sampled unrelated individuals of either European ancestry (for cross-ancestry and European-specific GWAS) or AMR ancestry (for AMR-specific GWAS). The standard errors of the GWAS summary statistics were adjusted for the LD score regression intercept (LD score regression analysis) before GCTA-COJO analysis. We defined GWAS loci as 'known' if the index variant in the loci was in LD ($R^2 > 0.1$) with genome-wide significant variants reported previously[16]. LD calculations were carried out using PLINK version 2 (ref. [78]). Our list of known GWAS loci came primarily from Liu et al.[16]. However, before declaring a variant as 'new', we also manually queried the variants in the GWAS Catalog to ensure that the variants were not in LD with variants reported in other smoking GWAS publications.

## LD score regression analysis

We calculated SNP-$h^2$, that is, the proportion of phenotypic variance explained by the common variants, using LD score regression software[43]. We used a European LD reference panel built in house using a random set of 10,000 unrelated European individuals from the UKB following the instructions provided by the authors of the LD score regression software. Genetic correlations were also computed using LD score regression software using the European LD reference panel. We used LD score regression also to quantify the population stratification that is known to inflate GWAS association statistics[43]. We computed LD score intercepts for all GWAS runs including the cross-ancestry and AMR-specific GWAS and then compared the values to the corresponding genomic control (GC) $\lambda$ values. A GC $\lambda$ > 1 but an intercept = 1 suggests that the observed inflation in the test statistics is fully due to polygenicity. For phenotypes such as smoking that are substantially influenced by environmental factors, it is common to have intercept values slightly above 1 (but still lower than GC $\lambda$), indicating that there is inflation in test statistics due to factors other than polygenicity, for example, population stratification, cryptic relatedness, etc.[43]. To remove such inflation, we applied a correction factor[79] to the test statistics to constrain the LD score intercept close to 1. We scaled the standard errors of the variant associations by a factor of the square root of the LD score intercept. This is a better alternative to GC correction (commonly practiced in large-scale consortium GWAS), as GC correction tends to overcorrect the statistics, removing true polygenic signals[79]. The LD score statistics before and after intercept correction are reported in Supplementary Table 15. We used the European LD reference panel even for cross-ancestry as well as AMR-specific GWAS, as there are no well-established guidelines on how to handle cross-ancestry or admixed ancestry-based GWAS results. We acknowledge that this has likely biased the results toward variants that are shared between European and other ancestries.

## Polygenic score analysis

We calculated smoking PGS for the UKB participants using SNP weights based on a GWAS of the ever smoker phenotype conducted in an independent sample. We obtained the summary statistics of the most recent GWAS of the ever smoker phenotype from the GSCAN consortium based on an analysis of all the participating GSCAN cohorts except the UKB and 23andMe[19]. To improve the statistical power of the PGS,

we meta-analyzed the GSCAN results with the GWAS results of the GHS cohort, which together yielded a total sample size of 482,096 individuals. We then refined the SNP effect sizes in the GWAS summary statistics using PRS-CS software[80], which uses a Bayesian approach to calculate SNP posterior effect sizes under continuous shrinkage priors based on an external LD reference panel. The refined SNP weights are then used to compute PGS using PLINK version 2 software[78].

We performed two types of analysis. First, we studied the associations of PGS and *CHRNB2* pLOF-plus-missense burden with heavy smoking using logistic regression analysis, in which the heavy smoker phenotype was coded as the dependent variable (that is, outcome), and PGS, burden mask, an interaction term between PGS and burden mask and relevant covariates (the same as the ones used in the GWAS) were coded as independent variables (regression formula: heavy smoker ≈ PGS + burden mask + PGS × burden mask + covariate$_1$ + … covariate$_n$). Second, we binned UKB individuals into quintiles (five equally sized groups) based on their smoking PGS. Individuals within each quintile were further divided into carriers and non-carriers of *CHRNB2* pLOF or likely deleterious missense variants at MAF < 0.001. The prevalence of heavy smokers was then compared between carriers and non-carriers within each quintile; the standard error was calculated using the formula $\sqrt{(pq/n)}$, where $n$ is the number of individuals in the group, $p$ is the prevalence of heavy smokers in the group and $q$ is $1 - p$. We also tested the statistical difference in the prevalence of heavy smokers between carriers and non-carriers of rare variant burden using logistic regression analysis adjusted for relevant covariates (the same as the ones used in the GWAS). The OR, 95% CI and the $P$ value for each quintile are reported in Fig. 7.

## Power calculations

All power calculations were carried out in R using the package 'genpwr' available from CRAN[81]. In all cases, we computed effect sizes ($\beta$ values) using the function 'genpwr.calc' with the following input parameters: power = 0.80, calc = 'es', model = 'logistic' for binary phenotypes and 'linear' for quantitative phenotypes, $\alpha$ = '5 × 10$^{-8}$' for GWAS and '4.5 × 10$^{-8}$' for ExWAS, MAF = values ranging from 0 to 0.5, True. model = 'additive' and Test.model = 'additive', $n$ = total sample size, case_rate = $n$ cases($n$ total)$^{-1}$ (for binary phenotypes) and sd_y = 1 (for quantitative phenotypes).

## Reporting summary

Further information on research design is available in the Nature Portfolio Reporting Summary linked to this article.

# Data availability

The data supporting the findings of this study are reported in the main text, figures and Supplementary Tables 1–17. UKB individual-level genotypic and phenotypic data are available to approved investigators via the UKB study (https://www.ukbiobank.ac.uk/). Additional information about registration for access to the data is available at https://www.ukbiobank.ac.uk/register-apply/. Data access for approved applications requires a data-transfer agreement between the researcher's institution and the UKB, the terms of which are available on the UKB website (https://www.ukbiobank.ac.uk/media/ezrderzw/applicant-mta.pdf). GHS individual-level data are available to qualified academic noncommercial researchers through the portal at https://regeneron.envisionpharma.com/vt_regeneron/ under a data-access agreement. The MCPS represents a long-standing collaboration between researchers at the UNAM and the University of Oxford. The investigators welcome requests from researchers in Mexico and elsewhere who wish to access MCPS data. If you are interested in obtaining data from the study for research purposes or in collaborating with MCPS investigators on a specific research proposal, please visit https://www.ctsu.ox.ac.uk/research/mcps, where you can download the study's Data and Sample Access Policy in English or Spanish. The policy lists the data available

for sharing with researchers in Mexico and in other parts of the world. Full details of the available data may also be viewed at https://data-share.ndph.ox.ac.uk/. FinnGen release 7 genetic association results, which were used in the current study, are publicly available at https://r7.finngen.fi/.

## Code availability

All genetic association analyses were performed using REGENIE software version 2.0.1, developed in house. REGENIE software is freely available on GitHub (https://github.com/rgcgithub/regenie) and Zenodo (https://doi.org/10.5281/zenodo.6789126)[82].

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

## Acknowledgements

We thank the UKB team, their funders, the dedicated professionals from the member institutions who contributed to and supported this work and the UKB participants. Exome sequencing was funded by the UKB Exome Sequencing Consortium (that is, Bristol Myers Squibb, Regeneron, Biogen, Takeda, AbbVie, Alnylam, AstraZeneca and Pfizer). This research has been conducted using the UKB resource under application number 2604. We thank MyCode Community Health Initiative participants for taking part in the DiscovEHR collaboration. This research received funding from Regeneron Pharmaceuticals. We thank the participants of the MCPS cohort. The MCPS has received funding from the Mexican Health Ministry, the National Council of Science and Technology for Mexico, the Wellcome Trust (grant number 058299/Z/99; recipients: J.A., R.T.-C., P.K.M. and R. Collins), Cancer Research UK, the British Heart Foundation and the UK Medical Research Council (grant number MC_UU_00017/2; recipient: J.E.). These funding sources had no role in the design, conduct or analysis of the study or the decision to submit the manuscript for publication. Genotyping, exome sequencing and whole-genome sequencing were funded through an academic partnership between the UNAM, the University of Oxford, Regeneron, AstraZeneca and AbbVie. The computational aspects of this research were supported by the Wellcome Trust Core Award (grant number 203141/Z/16/Z) and the NIHR Oxford BRC. The views expressed are those of the authors and not necessarily those of the NHS, the NIHR or the UK Department of Health. We thank the participants and investigators of the FinnGen study. We thank G. Saunders and others from the GSCAN consortium for providing a version of GWAS summary statistics based on a GWAS that excluded the UKB. We thank S. Croll for the helpful discussions on the results of this study.

## Author contributions

All authors reviewed the manuscript for important intellectual content and approved the manuscript submitted for publication. Conceptualization: V. M. Rajagopal, A. Baras, G.C. Genetic analysis: V. M. Rajagopal, K.W., J. Mbatchou, A. Ayer, M.D.K., K. Praveen, S. Gelfman, N. Parikshak, J.M.O., S. Bao. Phenotype preparation and harmonization: D.S., M.C. Statistical method development: J. Mbatchou, J. Marchini, G.R.A. Analytical pipeline development: V. M. Rajagopal, K.W., J. Mbatchou, M.D.K., S. Bao, S. Balasubramanian, G.R.A., H.M.K., J. Marchini, E.A.S., E.J. Data curation: V. M. Rajagopal, K.W., J. Mbatchou, A. Ayer, P.Q., D.S., M.D.K., K. Praveen, S. Gelfman, N. Parikshak, J.M.O., S. Bao, S.M.C., E.P., M. Kapoor, J.E., R. Collins, J.T., P.K.M., R.T.-C., J.A., J. Berumen, A.R.S., S. Balasubramanian, G.R.A., H.M.K., J. Marchini, E.A.S., E.J., R. Sanchez, M.C., D. Lederer, A. Baras, G.C. Funding acquisition: A. Baras, J.E., R. Collins, P.K.M., R.T.-C., J.A., J. Berumen. Project administration: M.B.J., M. Leblanc, E.C. Supervision: A. Baras, G.C., D. Lederer, M.C., M.A., W.L., E.J., E.A.S., J. Marchini, H.M.K., G.R.A., S. Balasubramanian, A.R.S. Writing (original draft): V. M. Rajagopal, G.C. All authors contributed to securing funding, study design and oversight; reviewed the final version of the manuscript; performed and were responsible for sample genotyping and exome sequencing; conceived and were responsible for laboratory automation, sample tracking and the library information-management system; were responsible for development and validation of the clinical phenotypes used to identify study participants and (when applicable) controls; performed and were responsible for the analysis needed to produce exome and genotype data; provided computing infrastructure development and operational support; provided variant and gene annotations and their functional interpretation of variants and conceived and were responsible for creating, developing and deploying the analysis platforms and

computational methods used to analyze the genomic data; developed the statistical analysis plans; contributed to quality control of the genotype and phenotype files and the generation of the analysis-ready datasets; developed the statistical genetic pipelines and tools and use thereof in the generation of association results; contributed to quality control of the review and interpretation of results and generated and formatted the results to create the manuscript figures; contributed to development of the study design and analysis plans and quality control of phenotype definitions; quality controlled, reviewed and interpreted the association results; contributed to the management and coordination of all research activities, planning and execution and managed the review of the project.

## Competing interests

V. M. Rajagopal, K.W., J. Mbatchou, A. Ayer, P.Q., D.S., M.D.K., K. Praveen, S. Gelfman, N. Parikshak, J.M.O., S. Bao, S.M.C., E.P., A. Avbersek, M. Kapoor, E.C., M.B.J., M. Leblanc, A.R.S., S. Balasubramanian, G.R.A., H.M.K., J. Marchini, E.A.S., E.J., R. Sanchez, W.L., M.A., M.C., D. Lederer, A. Baras and G.C. are current or former employees and/or stockholders of Regeneron Pharmaceuticals. The other authors declare no competing interests.

## Additional information

**Extended data** is available for this paper at https://doi.org/10.1038/s41588-023-01417-8.

**Correspondence and requests for materials** should be addressed to Aris Baras or Giovanni Coppola.

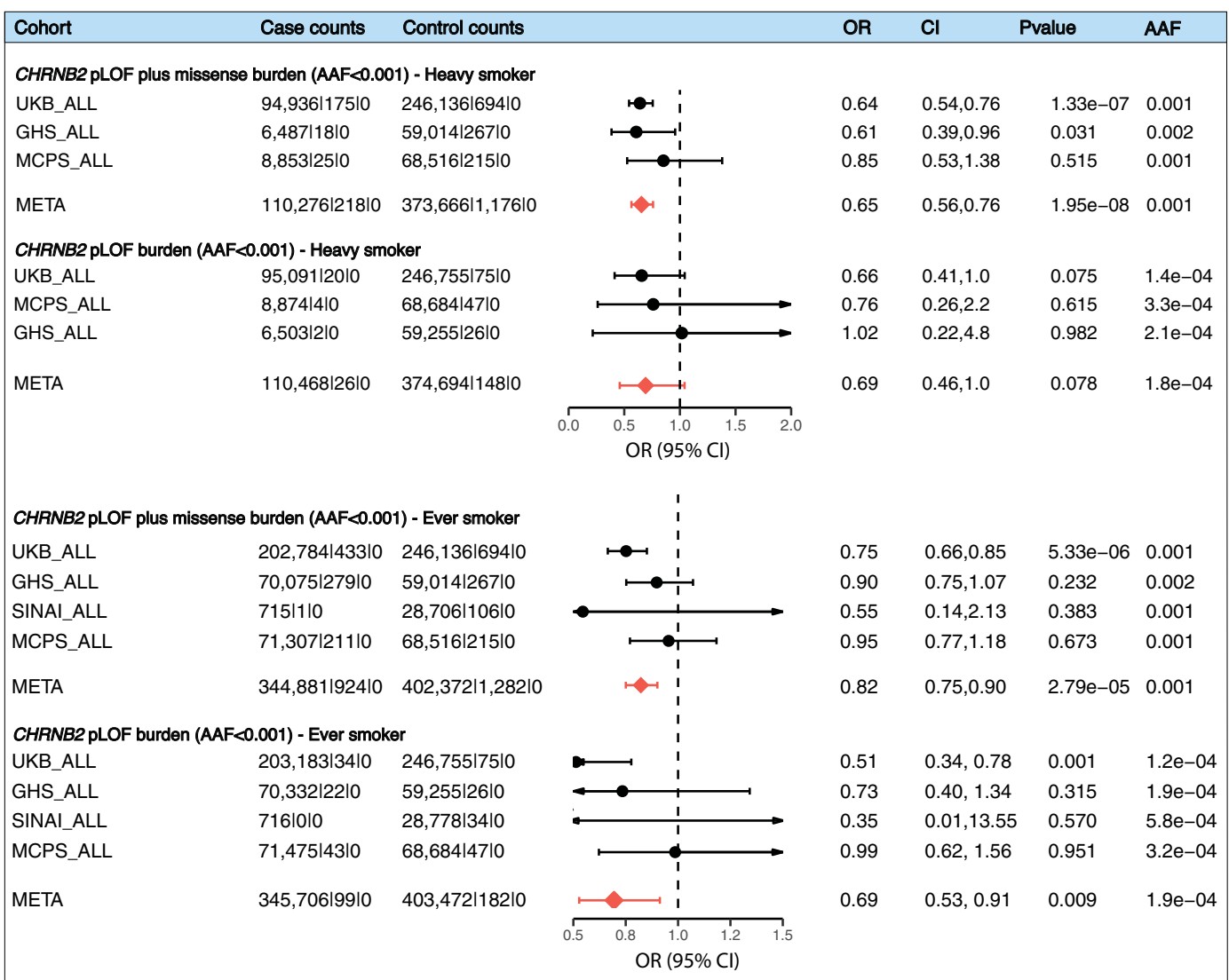

| Cohort | Case counts | Control counts | | OR | CI | Pvalue | AAF |
|---|---|---|---|---|---|---|---|
| **CHRNB2 pLOF plus missense burden (AAF<0.001) - Heavy smoker** | | | | | | | |
| UKB_ALL | 94,936\|175\|0 | 246,136\|694\|0 | | 0.64 | 0.54,0.76 | 1.33e−07 | 0.001 |
| GHS_ALL | 6,487\|18\|0 | 59,014\|267\|0 | | 0.61 | 0.39,0.96 | 0.031 | 0.002 |
| MCPS_ALL | 8,853\|25\|0 | 68,516\|215\|0 | | 0.85 | 0.53,1.38 | 0.515 | 0.001 |
| META | 110,276\|218\|0 | 373,666\|1,176\|0 | | 0.65 | 0.56,0.76 | 1.95e−08 | 0.001 |
| **CHRNB2 pLOF burden (AAF<0.001) - Heavy smoker** | | | | | | | |
| UKB_ALL | 95,091\|20\|0 | 246,755\|75\|0 | | 0.66 | 0.41,1.0 | 0.075 | 1.4e−04 |
| MCPS_ALL | 8,874\|4\|0 | 68,684\|47\|0 | | 0.76 | 0.26,2.2 | 0.615 | 3.3e−04 |
| GHS_ALL | 6,503\|2\|0 | 59,255\|26\|0 | | 1.02 | 0.22,4.8 | 0.982 | 2.1e−04 |
| META | 110,468\|26\|0 | 374,694\|148\|0 | | 0.69 | 0.46,1.0 | 0.078 | 1.8e−04 |
| **CHRNB2 pLOF plus missense burden (AAF<0.001) - Ever smoker** | | | | | | | |
| UKB_ALL | 202,784\|433\|0 | 246,136\|694\|0 | | 0.75 | 0.66,0.85 | 5.33e−06 | 0.001 |
| GHS_ALL | 70,075\|279\|0 | 59,014\|267\|0 | | 0.90 | 0.75,1.07 | 0.232 | 0.002 |
| SINAI_ALL | 715\|1\|0 | 28,706\|106\|0 | | 0.55 | 0.14,2.13 | 0.383 | 0.001 |
| MCPS_ALL | 71,307\|211\|0 | 68,516\|215\|0 | | 0.95 | 0.77,1.18 | 0.673 | 0.001 |
| META | 344,881\|924\|0 | 402,372\|1,282\|0 | | 0.82 | 0.75,0.90 | 2.79e−05 | 0.001 |
| **CHRNB2 pLOF burden (AAF<0.001) - Ever smoker** | | | | | | | |
| UKB_ALL | 203,183\|34\|0 | 246,755\|75\|0 | | 0.51 | 0.34, 0.78 | 0.001 | 1.2e−04 |
| GHS_ALL | 70,332\|22\|0 | 59,255\|26\|0 | | 0.73 | 0.40, 1.34 | 0.315 | 1.9e−04 |
| SINAI_ALL | 716\|0\|0 | 28,778\|34\|0 | | 0.35 | 0.01,13.55 | 0.570 | 5.8e−04 |
| MCPS_ALL | 71,475\|43\|0 | 68,684\|47\|0 | | 0.99 | 0.62, 1.56 | 0.951 | 3.2e−04 |
| META | 345,706\|99\|0 | 403,472\|182\|0 | | 0.69 | 0.53, 0.91 | 0.009 | 1.9e−04 |

**Extended Data Fig. 1 | Forest plots of CHRNB2 burden associations with heavy-smoker and ever-smoker.** The forest plot displays the cohort-level and meta-analysis associations of the CHRNB2 pLOF-only (AAF<0.001) and pLOF plus missense (AAF<0.001) burden masks with heavy-smoker and ever-smoker tested using REGENIE (Methods). The odds ratios and 95% confidence intervals are plotted. The columns 'case counts' and 'control counts' show the case and control sample sizes, respectively, broken down to the number of carriers of the homozygous reference, heterozygous and homozygous alternative genotypes.

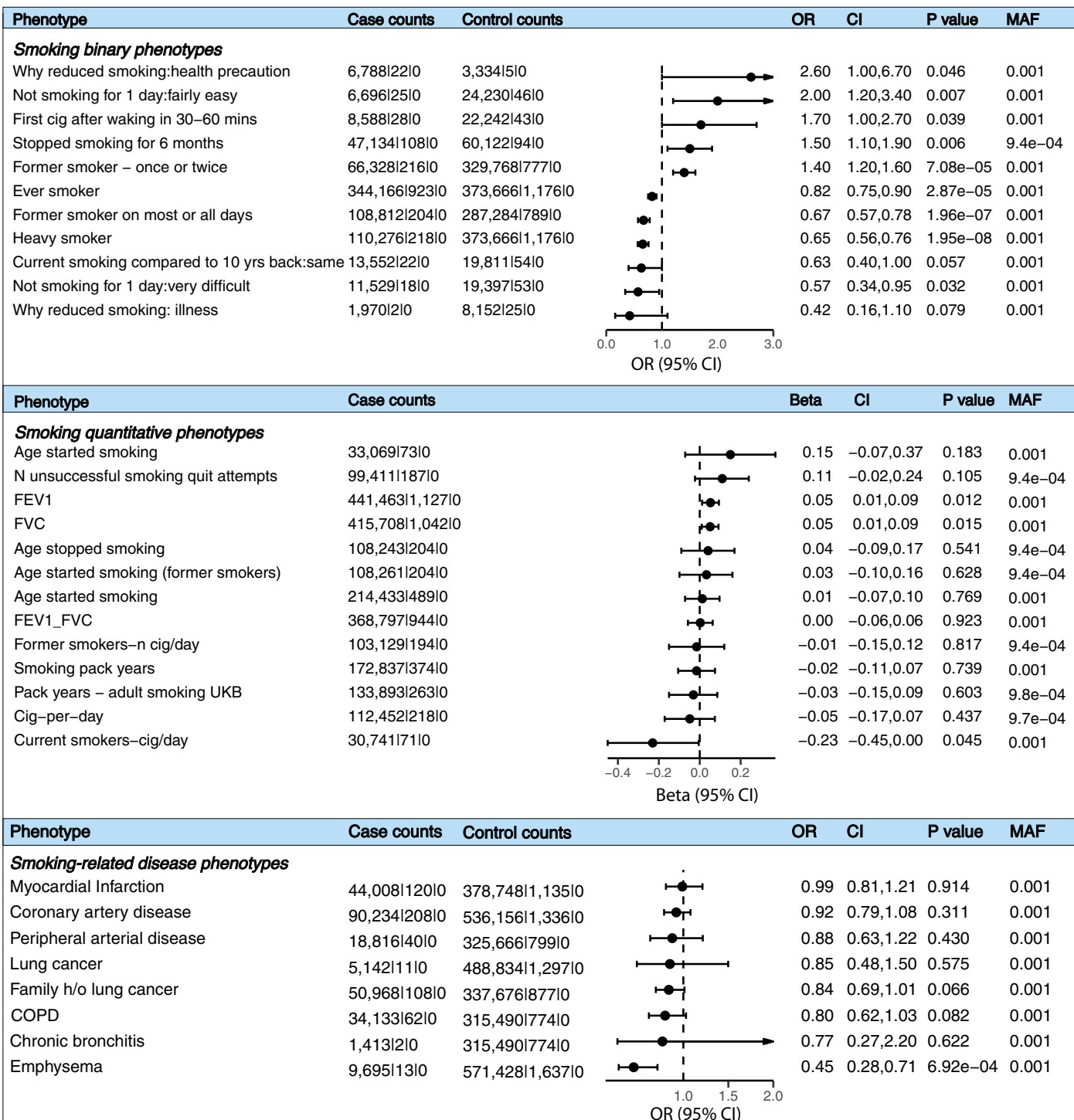

**Extended Data Fig. 2 | Forest plots of *CHRNB2* burden associations with secondary smoking phenotypes.** The forest plots display the cohort-level or meta-analysis associations of *CHRNB2* pLOF plus missense (AAF<0.001) burden mask with binary (P<0.1) and quantitative smoking phenotypes (major smoking phenotypes and phenotypes derived based on UKB lifestyle questionnaire) and smoking-related diseases tested using REGENIE (Methods). The odds ratios (or Beta estimates) and 95% confidence intervals are plotted. The columns 'case counts' and 'control counts' show the case and control sample sizes, respectively, broken down to the number of carriers of the homozygous reference, heterozygous and homozygous alternative genotypes. FEV1 – Forced expiratory volume in 1 sec; FVC – Forced vital capacity; FEV1_FVC – FEV1:FVC ratio; COPD – Chronic obstructive pulmonary disease.

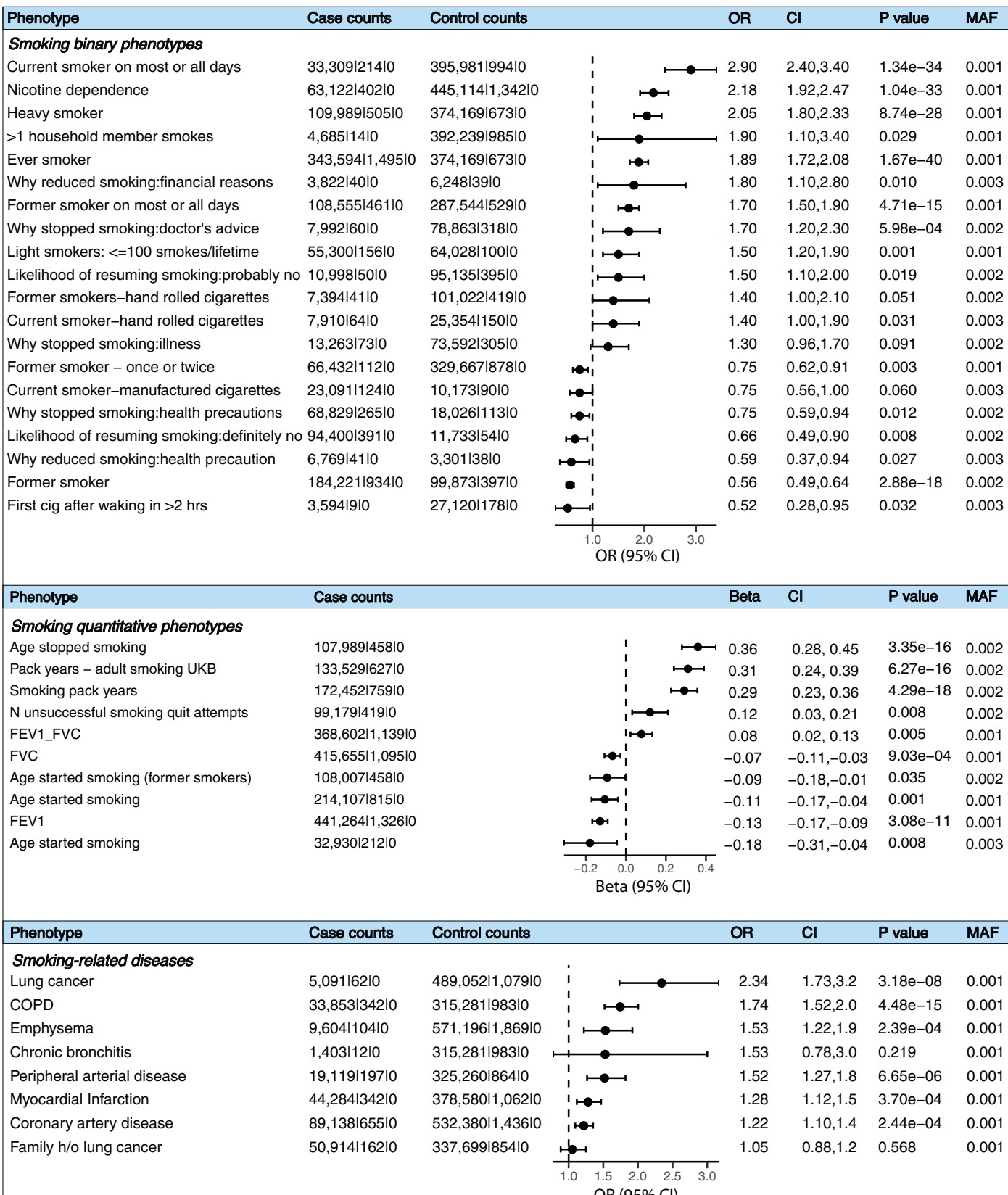

**Extended Data Fig. 3 | Forest plots of *ASXL1* burden associations with secondary smoking phenotypes.** The forest plots display the cohort-level or meta-analysis associations of *ASXL1* pLOF only burden mask (AAF<0.01) with binary and quantitative smoking phenotypes (major smoking phenotypes and phenotypes derived based on UKB lifestyle questionnaire with P<0.1) and smoking-related diseases tested using REGENIE (Methods). The odds ratios (or beta estimates) and 95% confidence intervals are plotted. The columns 'case counts' and 'control counts' show the case and control sample sizes, respectively, broken down to the number of carriers of the homozygous reference, heterozygous and homozygous alternative genotypes.

| Phenotype | Case counts | Control counts | | OR | CI | P value | MAF |
|---|---|---|---|---|---|---|---|
| *Smoking binary phenotypes* | | | | | | | |
| Likelihood of resuming smoking: probably yes | 1,406\|26\|0 | 103,678\|1,468\|0 | | 1.70 | 1.10,2.60 | 0.015 | 0.007 |
| Current smoker on most or all days | 33,055\|468\|0 | 392,231\|4,744\|0 | | 1.40 | 1.20,1.50 | 3.02e−09 | 0.006 |
| Former smokers–cigars/pipes | 4,946\|93\|0 | 102,400\|1,437\|0 | | 1.30 | 1.00,1.60 | 0.019 | 0.007 |
| Current smoker–manufactured cigarettes | 22,870\|345\|0 | 10,141\|122\|0 | | 1.30 | 1.00,1.60 | 0.037 | 0.007 |
| Current smoking compared to 10 yrs back:less | 14,524\|244\|0 | 18,447\|224\|0 | | 1.30 | 1.00,1.50 | 0.016 | 0.007 |
| Nicotine dependence | 64,914\|903\|0 | 467,683\|5,901\|0 | | 1.28 | 1.19,1.38 | 2.04e−10 | 0.006 |
| Ever smoker | 341,492\|4,311\|0 | 399,437\|4,190\|0 | | 1.19 | 1.14,1.25 | 4.81e−14 | 0.005 |
| Heavy smoker | 109,005\|1,489\|0 | 370,993\|3,849\|0 | | 1.18 | 1.10,1.26 | 9.04e−07 | 0.005 |
| Former smoker on most or all days | 107,483\|1,533\|0 | 284,858\|3,215\|0 | | 1.10 | 1.10,1.20 | 3.54e−05 | 0.006 |
| Former smoker | 182,543\|2,612\|0 | 99,325\|945\|0 | | 0.84 | 0.77,0.91 | 1.84e−05 | 0.006 |
| Former smokers–manufactured cigarettes | 95,571\|1,353\|0 | 11,775\|177\|0 | | 0.81 | 0.69,0.96 | 0.016 | 0.007 |
| Current smoking compared to 10 yrs back:more | 5,499\|52\|0 | 27,472\|416\|0 | | 0.71 | 0.53,0.95 | 0.020 | 0.007 |

OR (95% CI)

| Phenotype | Case counts | | Beta | CI | P value | MAF |
|---|---|---|---|---|---|---|
| *Smoking quantitative phenotypes* | | | | | | |
| Age stopped smoking | 106,926\|1,521\|0 | | 0.11 | 0.06,0.16 | 5.98e−06 | 0.007 |
| Pack years – adult smoking UKB | 132,299\|1,857\|0 | | 0.09 | 0.04,0.13 | 1.59e−04 | 0.006 |
| Smoking pack years | 171,016\|2,195\|0 | | 0.08 | 0.04,0.11 | 1.52e−04 | 0.006 |
| FEV1_FVC | 365,167\|4,574\|0 | | 0.04 | 0.01,0.06 | 0.010 | 0.006 |
| FEV1 | 437,203\|5,387\|0 | | −0.02 | −0.04,0.00 | 0.099 | 0.006 |

Beta (95% CI)

| Phenotype | Case counts | Control counts | | OR | CI | P value | MAF |
|---|---|---|---|---|---|---|---|
| *Smoking-related disease phenotypes* | | | | | | | |
| Lung cancer | 5,020\|133\|0 | 485,556\|4,575\|0 | | 1.56 | 1.29,1.9 | 7.14e−06 | 0.004 |
| COPD | 33,966\|663\|0 | 316,966\|4,145\|0 | | 1.07 | 0.98,1.2 | 0.149 | 0.006 |
| Emphysema | 9,901\|228\|0 | 594,539\|7,763\|0 | | 1.06 | 0.92,1.2 | 0.418 | 0.006 |
| Family h/o lung cancer | 50,406\|670\|0 | 334,725\|3,828\|0 | | 1.04 | 0.95,1.1 | 0.422 | 0.005 |
| Peripheral arterial disease | 20,408\|425\|0 | 346,295\|4,090\|0 | | 1.00 | 0.90,1.1 | 0.943 | 0.006 |
| Coronary artery disease | 93,105\|1,638\|0 | 554,572\|6,602\|0 | | 0.97 | 0.91,1.0 | 0.328 | 0.006 |
| Chronic bronchitis | 1,468\|26\|0 | 316,967\|4,145\|0 | | 0.97 | 0.66,1.4 | 0.856 | 0.006 |
| Myocardial Infarction | 44,625\|809\|0 | 399,465\|4,937\|0 | | 0.96 | 0.88,1.0 | 0.295 | 0.006 |

OR (95% CI)

**Extended Data Fig. 4 | Forest plots of *DNMT3A* burden associations with secondary smoking phenotypes.** The forest plots display the cohort-level or meta-analysis associations of *DNMT3A* pLOF plus missense burden mask (AAF<0.01) with binary and quantitative smoking phenotypes (major smoking phenotypes and phenotypes derived based on the UKB lifestyle questionnaire) and smoking-related diseases tested using REGENIE (Methods). The odds ratios (or beta estimates) and 95% confidence intervals are plotted. The columns 'case counts' and 'control counts' show the case and control sample sizes, respectively, broken down to the number of carriers of the homozygous reference, heterozygous and homozygous alternative genotypes.

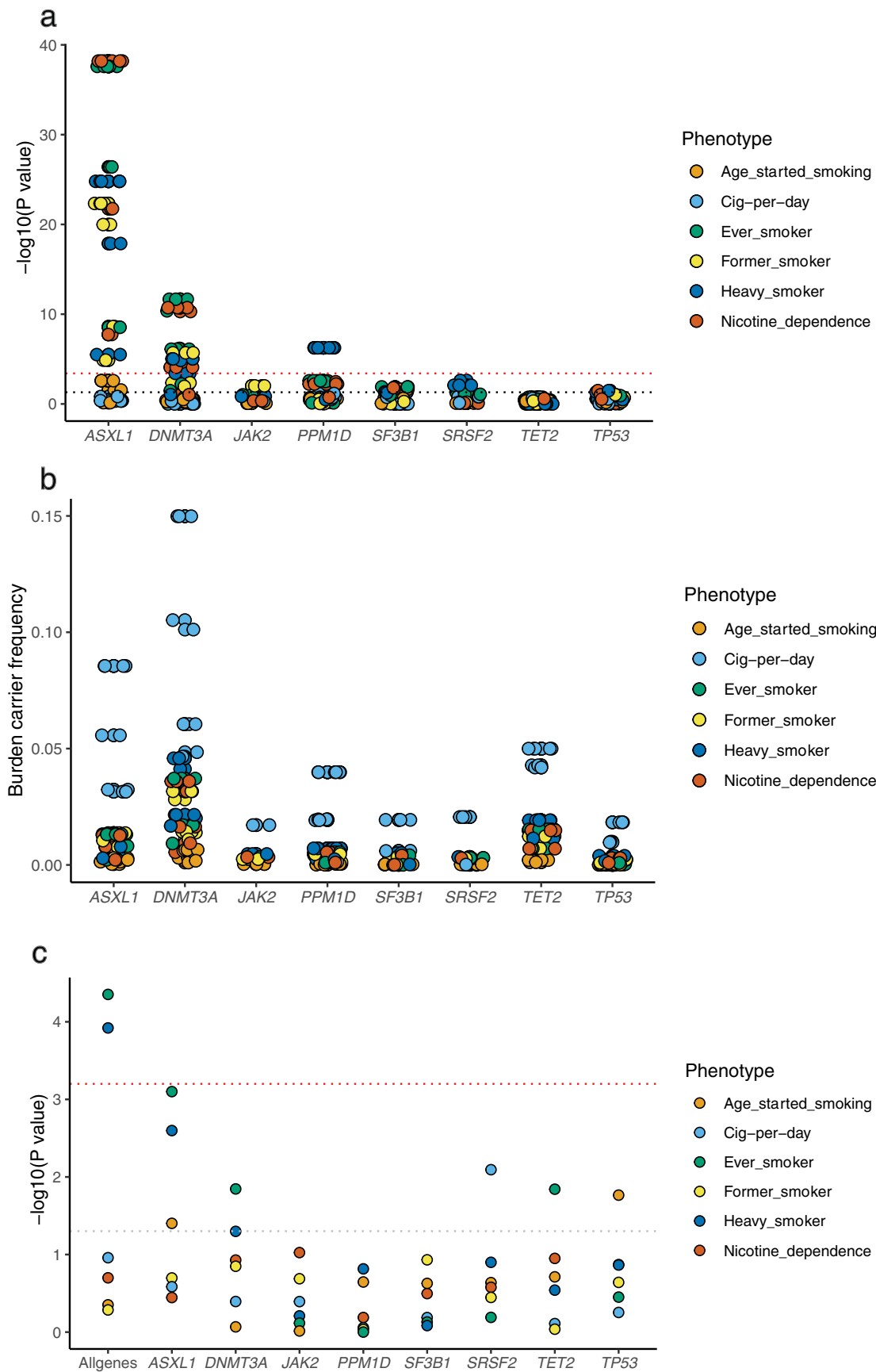

**Extended Data Fig. 5 | See next page for caption.**

**Extended Data Fig. 5 | Associations of CHIP mutations with smoking. a**. pLOF only and pLOF plus missense burden masks for eight recurrent CHIP genes were created in the UKB and GHS cohorts by aggregating only high-confident CHIP mutations (Methods) and tested for their associations with the six smoking phenotypes. The results were meta-analyzed between the GHS and UKB cohorts and the resulting P values are plotted. The dotted red line corresponds to FDR 1% P value threshold and the black dotted line corresponds to P = 0.05. **b**. The alternative allele frequencies (AAF) of the burden masks (combined AAF of all the variants aggregated in a mask) are plotted. **c**. Variant allele fractions (VAF) of CHIP mutations in the eight most recurrent CHIP genes were aggregated gene-wise and all together in the CHIP carriers in the UKB and GHS cohorts (when the same individual carried more than one CHIP mutation, we took the average of the VAF) and tested for associations with the six smoking phenotypes. The UKB and GHS combined association P values are plotted. The red dotted line corresponds to FDR 1% P value and the black dotted line corresponds to P = 0.05.

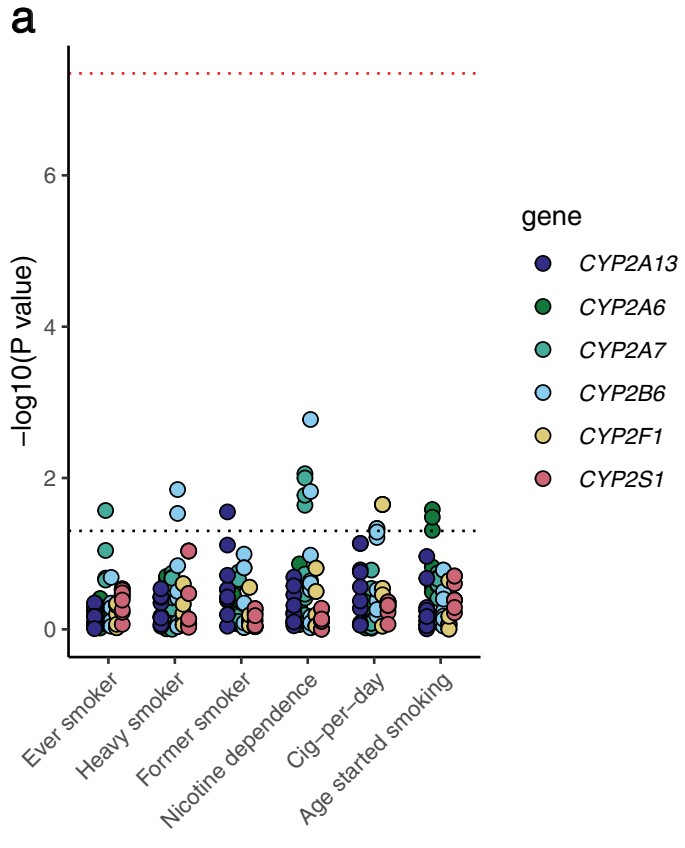

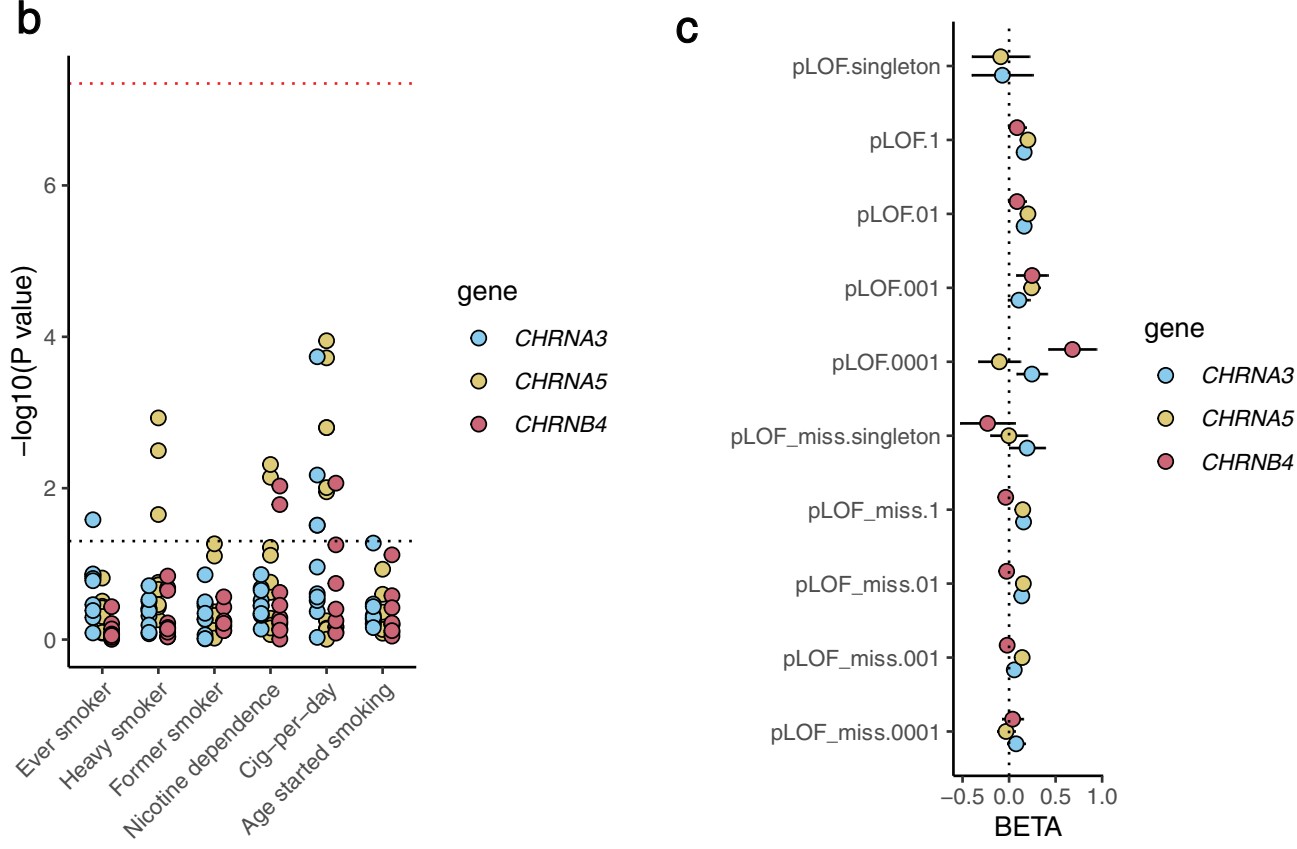

**Extended Data Fig. 6 | See next page for caption.**

**Extended Data Fig. 6 | Rare variant associations at the classic *CYP2A6* and *CHRNA5* GWAS loci. a**. P values of the pLOF only and pLOF plus missense burden associations of cytochrome gene cluster at the *CYP2A6* GWAS locus with the six smoking phenotypes are plotted. The red dotted line corresponds to FDR 1% P value and the black dotted line corresponds to P = 0.05. **b**. P values of the pLOF only and pLOF plus missense burden associations of nicotine acetylcholine receptor (nAChR) genes at the *CHRNA5* GWAS locus with the six smoking phenotypes are plotted. The red dotted line corresponds to FDR 1% P value and the black dotted line corresponds to P = 0.05. **c**. The beta estimates (in SD units) and 95% confidence intervals of the nAChR burden associations with cig-per-day (N = 112,670) are plotted. The sample sizes of the associations shown in panels a, b, and c are provided in Supplementary Table 11.

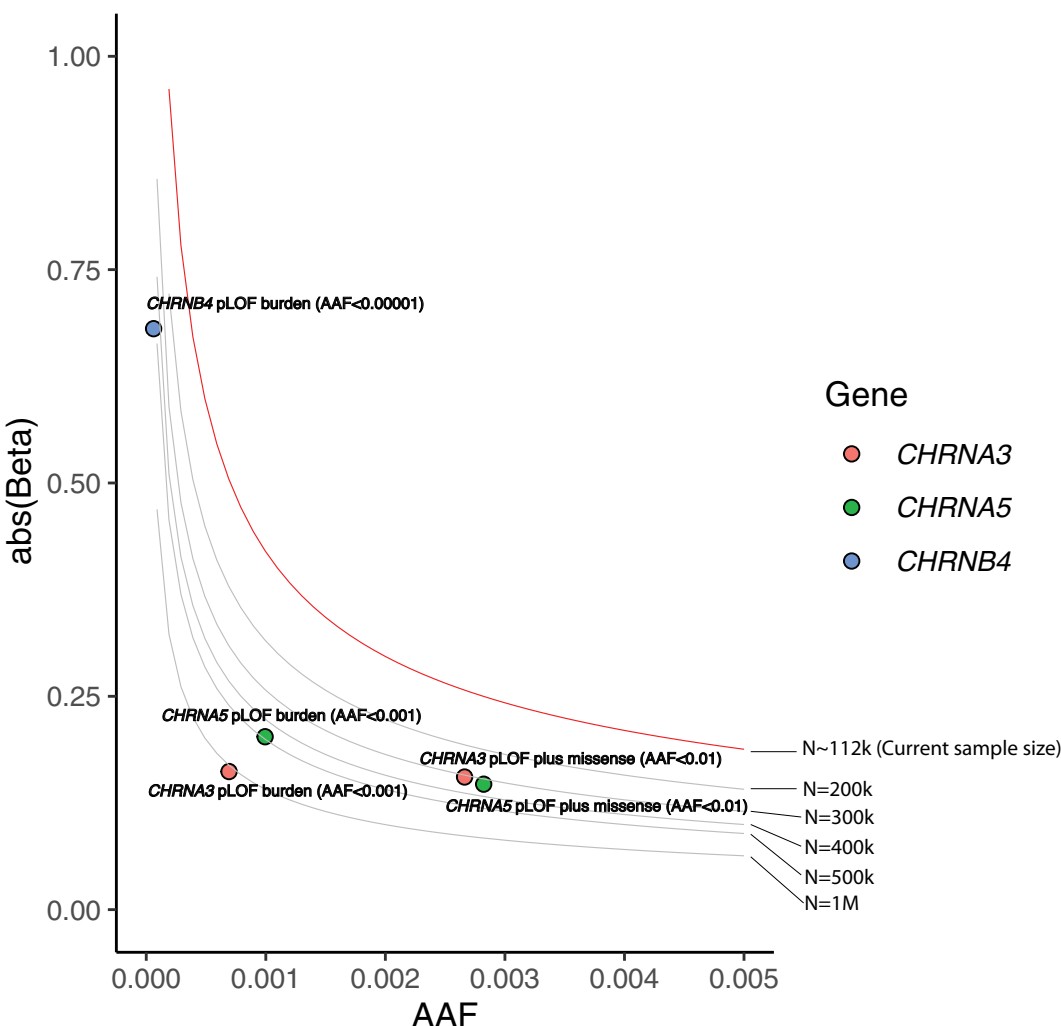

**Extended Data Fig. 7 | Power calculations for rare variant discovery at the _CHRNA5_ GWAS locus.** Assuming an 80% power and P value of 5e-8, detectable effect sizes at various minor allele frequency values were calculated for the current sample size of cig-per-day (the smoking trait most associated with _CHRNA5_ locus) as well for a series of sample sizes up to 1 million. The observed effect sizes for pLOF only burden and pLOF and missense burden associations of _CHRNA5_, _CHRNA3_ and _CHRNB4_ are plotted; all the points lay below the red line, which marks the detection limit of our current sample size, suggesting that we are underpowered. Based on the intersections of the grey lines with the points marking the observed effect sizes, we can approximately guess what sample size will be required to detect these burden signals at P value 5e-8.

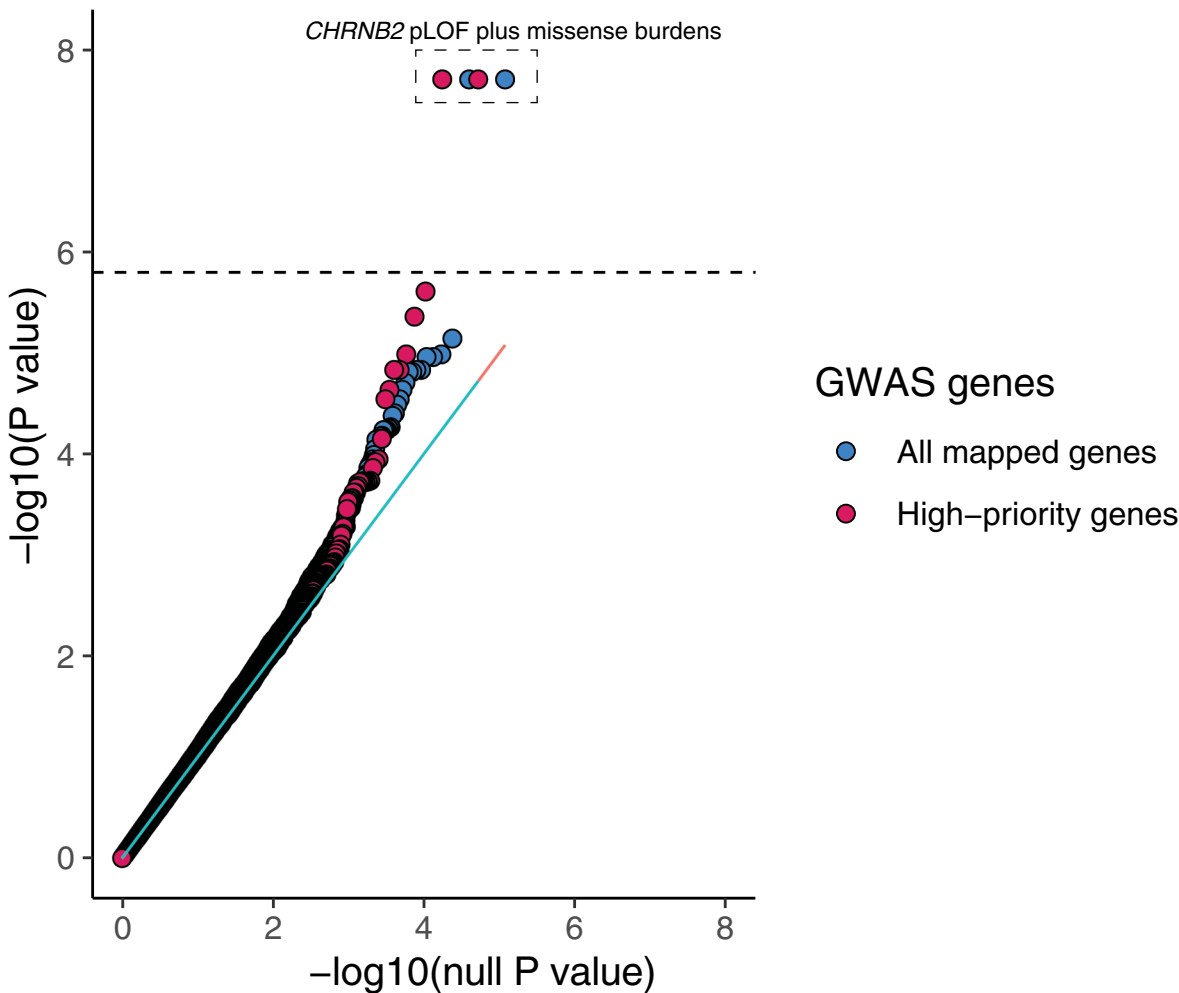

**Extended Data Fig. 8 | Association of rare variant burden in genes at the GWAS loci associated with smoking behavior.** Rare pLOF only and pLOF and missense burden associations were tested focusing only on the genes located at the known GWAS loci identified by the recent largest GWAS of smoking to date. We studied two gene lists prioritized by Saunders et al.[19]: a list of genes mapped to all the identified GWAS loci and a list of 'high-priority genes' mapped to GWAS loci with less than five fine-mapped variants. QQ plots of the meta-analysis P values of burden associations are shown. The dashed line corresponds to FDR 1% P value threshold.

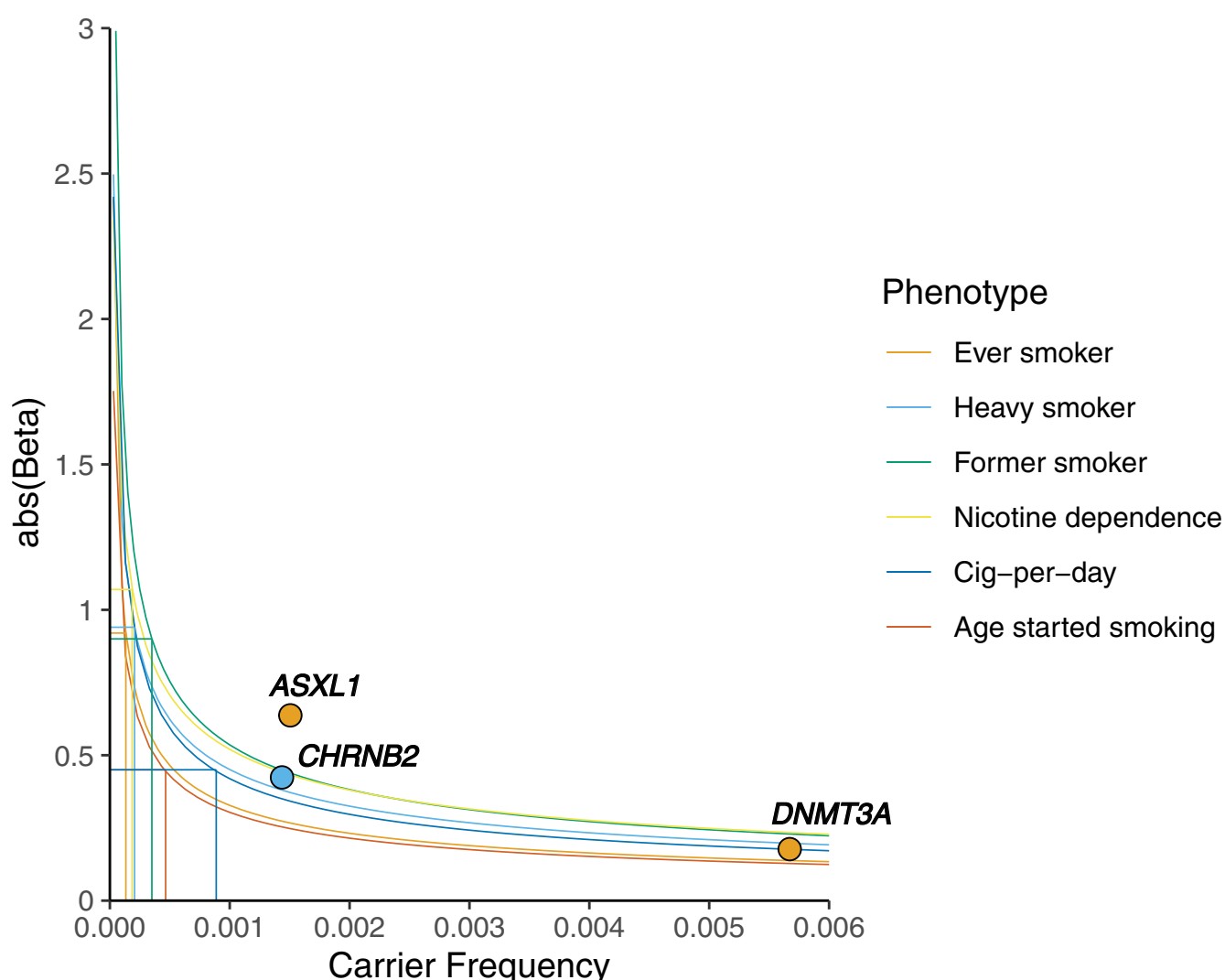

**Extended Data Fig. 9 | Power calculations for gene discovery using the current sample size.** Assuming an 80% power, P value threshold of 4e-8 (exome-wide significant threshold of the current study based on FDR 1%), effect sizes (that is, beta values) were computed for a range of minor allele frequencies (combined allele frequency in case of burden masks) for a given sample size (varies across phenotypes). The computed effect sizes (absolute values of beta estimates) are plotted against minor allele frequencies (carrier frequency) for six smoking phenotypes. The carrier frequency corresponding to 100 carriers, calculated for each of the phenotype based on the corresponding sample size, in the X axis and the corresponding effect size in the Y axis are marked with straight lines. The top association of the three genes identified as exome-wide significant are plotted with the color corresponding to the associated phenotype. Based on these power curves, we had 80% power to detect any variant or burden associations with ever-smoker, heavy-smoker and /former-smoker/ with odds ratio -2.5 or higher (0.4 or lower) when there are at least 100 carriers. And we had 80% power to detect any variant or burden associations with cig-per-day and age started smoking with beta 0.45 (equivalent to 4.7 extra cigarettes for cig-per-day and 1.9 yr earlier age for age started smoking) when there are at least 100 carriers. These calculations assume that there is no heterogeneity in the effect sizes across the cohorts, which is never the case for complex traits such as smoking. Hence, these estimates should be considered arbitrary. Importantly, the effect sizes for protective associations with binary phenotypes are likely overestimated due to imbalances in the case-control ratios.

# Reporting Summary

## Statistics

For all statistical analyses, confirm that the following items are present in the figure legend, table legend, main text, or Methods section.

| n/a | Confirmed | |
|---|---|---|
| ☐ | ☒ | The exact sample size (*n*) for each experimental group/condition, given as a discrete number and unit of measurement |
| ☐ | ☒ | A statement on whether measurements were taken from distinct samples or whether the same sample was measured repeatedly |
| ☐ | ☒ | The statistical test(s) used AND whether they are one- or two-sided *Only common tests should be described solely by name; describe more complex techniques in the Methods section.* |
| ☐ | ☒ | A description of all covariates tested |
| ☐ | ☒ | A description of any assumptions or corrections, such as tests of normality and adjustment for multiple comparisons |
| ☐ | ☒ | A full description of the statistical parameters including central tendency (e.g. means) or other basic estimates (e.g. regression coefficient) AND variation (e.g. standard deviation) or associated estimates of uncertainty (e.g. confidence intervals) |
| ☐ | ☒ | For null hypothesis testing, the test statistic (e.g. *F*, *t*, *r*) with confidence intervals, effect sizes, degrees of freedom and *P* value noted *Give P values as exact values whenever suitable.* |
| ☒ | ☐ | For Bayesian analysis, information on the choice of priors and Markov chain Monte Carlo settings |
| ☒ | ☐ | For hierarchical and complex designs, identification of the appropriate level for tests and full reporting of outcomes |
| ☐ | ☒ | Estimates of effect sizes (e.g. Cohen's *d*, Pearson's *r*), indicating how they were calculated |

*Our web collection on statistics for biologists contains articles on many of the points above.*

## Software and code

Policy information about availability of computer code

| Data collection | None |
|---|---|
| Data analysis | Software used for data analysis include Regenie (v.3.2.1), LDSC(v1.0.1), PRS-CS(v1.0.0), R (v4.1.0), GCTA (v1.91.7), SHAPEIT (v4.2.0), IMPUTE (v5), Mutect2 (GATK v4.1.4.0) |

For manuscripts utilizing custom algorithms or software that are central to the research but not yet described in published literature, software must be made available to editors and reviewers. We strongly encourage code deposition in a community repository (e.g. GitHub). See the Nature Portfolio guidelines for submitting code & software for further information.

## Data

Policy information about availability of data

All manuscripts must include a data availability statement. This statement should provide the following information, where applicable:
- Accession codes, unique identifiers, or web links for publicly available datasets
- A description of any restrictions on data availability
- For clinical datasets or third party data, please ensure that the statement adheres to our policy

UKB individual-level genotypic and phenotypic data are available to approved investigators via the UK Biobank study (www.ukbiobank.ac.uk/). Additional information about registration for access to the data are available at www.ukbiobank.ac.uk/register-apply/. Data access for approved applications requires a data transfer agreement between the researcher's institution and UK Biobank, the terms of which are available on the UK Biobank website (www.ukbiobank.ac.uk/

## Human research participants

Policy information about studies involving human research participants and Sex and Gender in Research.

| | |
|---|---|
| Reporting on sex and gender | Sex is included as a covariate in the genetic association analysis. Sex is inferred from the genetic data and was confirmed by comparing with the self reported sex. We did not use gender information for any of the analysis. |
| Population characteristics | Provided in the supplementary table 2 |
| Recruitment | Participant recruitment information for the respective cohorts is described in the methods section along with appropriate references. |
| Ethics oversight | All the study participants have provided informed consent and all the participating cohorts have received ethical approval from their respective institutional review board (IRB). The UK Biobank project has received ethical approval from the Northwest Centre for Research Ethics Committee (11/NW/0382). The work described here has been approved by the UKB (application no. 26041). The GHS project has received ethical approval from the Geisinger Health System Institutional Review Board under project no. 2006-025862. The MCPS study has received ethical approval from the Mexican Ministry of Health, the Mexican National Council for Science and Technology, and the University of Oxford. The BioMe biobank has received ethical approval from the IRB at the Icahn School of Medicine at Mount Sinai. |

Note that full information on the approval of the study protocol must also be provided in the manuscript.

# Field-specific reporting

Please select the one below that is the best fit for your research. If you are not sure, read the appropriate sections before making your selection.

☒ Life sciences          ☐ Behavioural & social sciences          ☐ Ecological, evolutionary & environmental sciences

For a reference copy of the document with all sections, see nature.com/documents/nr-reporting-summary-flat.pdf

# Life sciences study design

All studies must disclose on these points even when the disclosure is negative.

| | |
|---|---|
| Sample size | Sample size was not calculated prior to study. All samples available after quality control were included for analysis. |
| Data exclusions | Certain samples and genetic variants were excluded as part of the standard quality control pipeline applicable to any genetic association study. Details can be found in the methods and the cited references. |
| Replication | We did not have a separate replication cohort internally. We pooled genetic data from all our internal cohorts (UKB, GHS, MCPS and SINAI) to perform a meta-analysis. We identified three significant genes (ASXL1, DNMT3A and CHRNB2) for which we looked for consistency in the effect size directions and evidence for statistical significance (P<0.05) in the individual cohorts. The meta-analysis and individual cohort results of all three genes are reported in the manuscript. For all three genes, we observed a consistent direction of effect in at least three cohorts. and both a consistent direction of effect and statistical significance in at least two cohorts (Fig. 3). In addition, we replicated the protective association of a rare missense variant (Arg460Gly) with smoking-related phenotypes (substance use disorder and COPD) using the publicly available genetic association results from Finngen (release 7). |
| Randomization | Randomization is not applicable or possible in this study as it is a genetic association study based on hundreds of thousands of humans whose phenotypic information were collected retrospectively from the Electronic Health Records or health questionnaires responses of the participants or patients. |
| Blinding | Blinding is not required in this study as the phenotyping, genotyping and statistical analyses are completely independent processes and each happened without any prior knowledge of the others. |

# Reporting for specific materials, systems and methods

We require information from authors about some types of materials, experimental systems and methods used in many studies. Here, indicate whether each material, system or method listed is relevant to your study. If you are not sure if a list item applies to your research, read the appropriate section before selecting a response.

## Materials & experimental systems

| n/a | Involved in the study |
|-----|----------------------|
| ☒ ☐ | Antibodies |
| ☒ ☐ | Eukaryotic cell lines |
| ☒ ☐ | Palaeontology and archaeology |
| ☒ ☐ | Animals and other organisms |
| ☒ ☐ | Clinical data |
| ☒ ☐ | Dual use research of concern |

## Methods

| n/a | Involved in the study |
|-----|----------------------|
| ☒ ☐ | ChIP-seq |
| ☒ ☐ | Flow cytometry |
| ☒ ☐ | MRI-based neuroimaging |

