## [Peer Review File · Nature Genetics]

Peer Review Information

Manuscript Title: Rare coding variants in CHRN2 reduce the likelihood of smoking

Corresponding author name(s): Dr Giovanni Coppola and Dr Aris Baras

Reviewer Comments & Decisions:

Decision Letter, initial version:

14th Dec 2022

Dear Dr Coppola,

Your Article, "Rare coding variants in CHRN2 reduce the likelihood of smoking" has now been seen by 3 referees. You will see from their comments below that while they find your work of interest, some important points are raised. We are interested in the possibility of publishing your study in Nature Genetics, but would like to consider your response to these concerns in the form of a revised manuscript before we make a final decision on publication.

To guide the scope of the revisions, the editors discuss the referee reports in detail within the team with a view to identifying key priorities that should be addressed in revision. In this case, we think all three referees have provided constructive reviews aimed at strengthening the analyses and improving the presentation, and we particularly ask that you address their comments as thoroughly as possible with appropriate revisions. We hope that you will find the prioritized set of referee points to be useful when revising your study.

We therefore invite you to revise your manuscript taking into account all reviewer and editor comments. Please highlight all changes in the manuscript text file. At this stage we will need you to upload a copy of the manuscript in MS Word .docx or similar editable format.

*1) Include a “Response to referees” document detailing, point-by-point, how you addressed each referee comment. If no action was taken to address a point, you must provide a compelling argument. This response will be sent back to the referees along with the revised manuscript.

*2) If you have not done so already please begin to revise your manuscript so that it conforms to our Article format instructions, available [here](http://www.nature.com/ng/authors/article_types/index.html). Refer also to any guidelines provided in this letter.

[redacted]

We hope to receive your revised manuscript within three to six months. If you cannot send it within this time, please let us know.

Nature Genetics is committed to improving transparency in authorship. As part of our efforts in this direction, we are now requesting that all authors identified as ‘corresponding author’ on published papers create and link their Open Researcher and Contributor Identifier (ORCID) with their account on the Manuscript Tracking System (MTS), prior to acceptance. ORCID helps the scientific community achieve unambiguous attribution of all scholarly contributions. You can create and link your ORCID from the home page of the MTS by clicking on ‘Modify my Springer Nature account’. For more information please visit www.springernature.com/orcid.

Sincerely,
Wei

Wei Li, PhD
Senior Editor
Nature Genetics
New York, NY 10004, USA
www.nature.com/ng

Reviewers' Comments:

Reviewer #1:

Remarks to the Author:

Review. Rare coding variants in CHRNA2 reduce the likelihood of smoking

This is the first well powered rare variant association study of smoking behaviour. The study is very thorough; it includes different samples (including individuals from non-European ancestry) and six (primary) smoking phenotypes, it combines information on rare and common variants, replicates the main finding in an isolated population, and includes several follow-up analyses. Overall, the manuscript is written up well and has a good balance between describing findings and interpretations. The analytical approaches seem valid and the conclusions robust. The authors found a protective association of CHRNA2 rare variants (plus significant associations implicating 2 other genes), and they showed convergence of rare and common variant results in CHRNA2. This represents the first human genetic evidence supporting the hypothesis that loss of CHRNA2 protects people against nicotine dependence, and hence could be useful in discovering new therapeutics. The manuscript should be highly cited and fits well within the target journal.

I do not have any major comments, other than that the new GSCAN paper has just come out online with GWAS results from a substantially larger sample (>3M individuals), see <https://www.nature.com/articles/s41586-022-05477-4#MOESM3>. I think the authors will have to integrate these results in their manuscript and analyses (for example when calculating PGSs they should use the summary level data from this manuscript instead of the older GSCAN paper).

Minor comments:

- Page 3. '...with twin studies estimating its heritability up to 45%.' Heritability estimates for severe forms of smoking, e.g. nicotine dependence, are much higher than 45%; estimates are in the range of 70-85%. See for example <https://link.springer.com/article/10.1007/s10519-004-1327-8>.
- 'However, human genetic studies of smoking behavior have so far focused mainly on common variants (those observed in more than 1% of the population)^{15–17}. While this has been the case before, many GWASs do consider variants with a MAF below 1% (depending on the GWAS sample size). The GSCAN paper by Liu et al for example also looked at variants with a MAF between 0.1 and 1%, and in the new GSCAN paper that just came out (see above), they even included variants with a lower MAF than 0.1%.
- Methods: provide N when describing the different cohorts (page 11)

- Figure 1b: Why are there no data from Sinai for CHRNA2?
- Figure 2. Panel b has not been described in the legend.

Reviewer #2:

Remarks to the Author:

Summary of the key results

Rajagopal et al. performed an association study of multiple smoking phenotypes based on whole-exome sequencing data from multiple cohorts. They mainly focused on the rare variants in the coding regions and identified a signal in CHRNA2 in the aggregation analyses. They also presented data on common variants in this region and other reported regions. They proposed that CHIP mutations explained another two signals in ASXL1 and DNMT3A. This is an association study with large sample size. However, I have some concerns about the novelty of the results. In addition, some methods were not clearly described.

Originality and significance: if not novel, please include reference

1. The major finding of this study is a genetic signal in CHRNA2. However, the CHRNA2 region has been reported in a previous GWAS (PMID: 30643251), as mentioned by the authors. I agree with the authors that the rare coding variants may contribute greatly to the susceptibility of phenotypes, but the novelty of the exome-wide screening will be greatly diminished if the region has been reported.

2. Only CHIP mutations of ASXL1 and DNMT3A were associated with smoking behaviors; that is interesting. The authors propose their hypotheses (selective advantage); however, they do not present additional supporting data. They should look at the variant allele fraction (VAF) of these mutations significantly associated with smoking phenotypes.

Data & methodology: validity of approach, quality of data, quality of presentation

3. I do not see any methods for the analysis of CHIP mutations. How are these mutations defined? Also, I do not find the mean coverage of the whole-exome sequencing (not the percentage of 20X region); coverage might greatly influence the number of CHIP mutations. Is there any difference across different cohorts?

4. The signal of CHRN2 was associated with the phenotype of heavy smoker in UKB/GHS/MCPS. The readers may also want to know the association between CHRN2 variants and the phenotype of ever smoker in the SINAI study.

5. The authors provide a single P value for analyses based on multiple cohorts. What is the combined strategy? Direct combination or meta-analysis? If direct combination, the author may need to describe how to correct the batch effects. If meta-analysis, they may need to show the results of reported signals in each cohort.

6. Page 7, line 23. The effect of CHRN4 pLOF is different from that in Supplementary Table 10. What is the mean of “Beta=0.65 SD”?

Appropriate use of statistics and treatment of uncertainties

7. No statistical tests were performed in the ‘Interplay between common and rare variants’ section. Thus, any conclusions in this section may not be appropriate.

8. Page 5, line 33-34. No description of the methods for the enrichment analyses based on the data from the FinnGen project.

Conclusions: robustness, validity, reliability

9. Page 10, line 17-19. “However, to the best of our knowledge, what we describe here is the first human genetic evidence supporting the hypothesis that loss of CHRN2 protects against nicotine addiction.” This is not true. As mentioned by the authors themselves, Liu et al. have already presented the data of the CHRN2 region (rs2072659), as in Supplementary Table.

Suggested improvements: experiments, data for possible revision

10. Because smoking is the leading risk factor of lung cancer, the genetic variants associated with smoking commonly contribute to the risk of developing lung cancer (e.g., 15q25.1). Are rare CHRN2 variants associated with lung cancer risk in UKB data?

11. In previous GWAS, more than 400 loci were reported to be associated with smoking. What about the associations for the rare variants in these loci?

12. Heavy-smokers were defined as who smoked 10 or more cigarettes per day. Additional sensitivity analysis should be performed to test the robustness of the definition, as this definition was different from other studies, which used pack-years of 20/30 to define heavy smokers.

References: appropriate credit to previous work?

13. As whole-genome sequencing is gradually implemented in the population study (PMID: 32499645, 32581362, 33589841, 36113475), the authors may also discuss the limitation of the population study based on whole-exome sequencing.

Clarity and context: lucidity of abstract/summary, appropriateness of abstract, introduction and conclusions

14. Abstract. "α4β2 is the predominant nAChR in human brain and is one of the targets of varenicline, a partial nAChR agonist/antagonist used to aid smoking cessation." This is not the result of this association study. The description here is misleading.

Reviewer #3:

Remarks to the Author:

The manuscript by Rajagopal and colleagues describes a comprehensive ExWAS of smoking for rare variants in up to 749,459 individuals across multiple ancestries. The study leveraged off several well-known cohorts including UK biobank, GHS, MCPS, and SINAI. They used 6 primary smoking phenotypes for their analysis. The main finding is that rare coding variants in CHRNA2 is protective for smoking. The authors also identified two other genes, ASXL1, DNMT3A that reached ExWAS significance and determined that these genetic signals were mainly due to the CHIP variants. The relationship between common, as well as PGS, and rare variants of the CHRNA2 with smoking were also explored.

There is a prior publication from the TOPMED study that described the contribution of rare variants to smoking based on WGS (Jang et al, PMID: 35927319). As the authors stated, that study was not able to pinpoint to any specific gene. This study is novel and add to our current knowledge of genetics associated with smoking behavior by analyzing rare variants.

Their findings also supported by animal-based studies which showed protective effects of *CHRNA2* in nicotine dependence. Validation of a known 3'UTR SNP in *CHRNA2* with this study also support the validity of *CHRNA2* as a protective gene. This study supports *CHRNA2* as a potential therapeutic target for smoking cessation and addiction and the availability of known drugs targeting this important class of proteins.

Although the manuscript is well written, revision of the text to be more precise will improve the clarity and help the readers:

1. It probably better to move the supplemental fig 1 to the main text. The total N of subjects from each study included in this study should be indicated.
2. Add total N to the “Case counts” and “Control counts” for Fig 1b for each gene.
3. The section “Associations of rare and common variants in *CHRNA2*” in Result should be in two sections describing rare and common variants separately.
4. Long paragraph in the discussion maybe better in two paragraphs.

Author Rebuttal to Initial comments

Rare coding variants in *CHRNA2* reduce the likelihood of smoking

Revision summary

We thank the editors and reviewers for reviewing our manuscript and offering valuable comments and feedback. We have revised our manuscript as per the reviewers' recommendations. Some of the major revisions are summarized below:

Updated PGS analysis using new scores generated using the latest GWAS of smoking behavior by Saunders et al, Nature 2022.

Sensitivity analysis by defining heavy smoking in different ways and demonstrating the consistency of the protective association of *CHRNA2* rare variants with heavy smoking.

Association analysis variant allele fraction (VAF) of CHIP mutations with smoking phenotypes

Rare variant burden analysis focusing only on the known GWAS loci

Statistical tests for the PGS analysis

Below, we list a point-by-point response to reviewers' comments and a detailed description of the analysis and revisions we made to the manuscript.

Response to the reviewers' comments Reviewer #1:

This is the first well powered rare variant association study of smoking behaviour. The study is very thorough; it includes different samples (including individuals from non-European ancestry) and six (primary) smoking phenotypes, it combines information on rare and common variants, replicates the main finding in an isolated population, and includes several follow-up analyses. Overall, the manuscript is written up well and has a good balance between describing findings and interpretations. The analytical approaches seem valid and the conclusions robust. The authors found a protective association of CHRNA2 rare variants (plus significant associations implicating 2 other genes), and they showed convergence of rare and common variant results in CHRNA2. This represents the first human genetic evidence supporting the hypothesis that loss of CHRNA2 protects people against nicotine dependence, and hence could be useful in discovering new therapeutics. The manuscript should be highly cited and fits well within the target journal.

We thank the reviewer for this positive overview of our study.

I do not have any major comments, other than that the new GSCAN paper has just come out online with GWAS results from a substantially larger sample (>3M individuals), see <https://www.nature.com/articles/s41586-022-05477-4>. I think the authors will have to integrate these results in their manuscript and analyses (for example when calculating PGSs they should use the summary level data from this manuscript instead of the older GSCAN paper).

We thank the reviewer for this suggestion. We have now updated the PGS analysis using a new polygenic score that was based on larger training data (N~500k). We obtained the latest GSCAN summary statistics¹ (excluding UK Biobank and 23andme) and meta-analyzed with our GHS cohort to boost the sample size further (UKB cannot be included because it's the target sample, and MCPS cannot be included as it predominantly includes admixed Americans individuals). The new PGS yielded more statistical power in terms of visualizing the prevalence of heavy smokers across polygenic quintiles. We have further added new statistical analyses as per the request of Reviewer 2 (see our response to comment 15).

Minor comments:

- Page 3. '...with twin studies estimating its heritability up to 45%.' Heritability estimates for severe forms of smoking, e.g. nicotine dependence, are much higher than 45%; estimates are in the range of 70-85%. See for example <https://link.springer.com/article/10.1007/s10519-004-1327-8>.

We thank the reviewer for pointing to this reference. We agree that nicotine dependence has a higher twin heritability than smoking initiation. We have updated the text as below and updated the citation to Vink et al 2005².

"Smoking behavior is strongly influenced by genetics, with twin-based heritability estimates ranging between 45% (for smoking initiation) and 75% (for nicotine dependence)"

‘However, human genetic studies of smoking behavior have so far focused mainly on common variants (those observed in more than 1% of the population)^{15–17}.’ While this has been the case before, many GWASs do consider variants with a MAF below 1% (depending on the GWAS sample size). The GSCAN paper by Liu et al for example also looked at variants with a MAF between 0.1 and 1%, and in the new GSCAN paper that just came out (see above), they even included variants with a lower MAF than 0.1%.

We agree with the reviewer that the sentence about MAF threshold selection in previous GWASs needs clarity. In a GWAS, researchers often restrict the analysis to only variants with $MAF > 1\%$ as the imputation accuracy drops below that cut-off. However, with the recent increase in the GWAS sample sizes, researchers have started looking beyond MAF 1% cut-off, as the large sample size compensates for the loss of statistical power due to low imputation accuracy. For example, in the recent GWAS by Saunders et al.¹ the reviewer is referring to, the authors studied genetic variants with MAF up to 0.1% but chose only those variants with an effective sample size (actual sample size x imputation accuracy) of at least 10% of the study’s maximum sample size. Although this approach helps to study some variants with MAF between 0.1% to 1% with moderate imputation accuracy, it still misses many variants within this range that cannot be imputed confidently but can only be sequenced directly. It is challenging to describe these nuances to the reader briefly in the background section. At the same time, we agree with the reviewer that it’s not correct to bluntly say that past GWASs have focused only on common variants as that’s not entirely true. So, we have now updated the text below to add clarity on the MAF range we are referring to.

“... Genetic variants across the full minor allele frequency (MAF) spectrum—common ($MAF > 1\%$), low-frequency ($MAF 0.1\% - 1\%$), and rare variants ($MAF < 0.1\%$)—contribute to this high heritability. However, human genetic studies of smoking behavior have so far focused mainly on common and low-frequency variants (that can be imputed with at least a moderate accuracy). ...”

Methods: provide N when describing the different cohorts (page 11)

We have reported in detail the sample sizes for the six smoking phenotypes for individual cohorts and ancestries in Supplementary Table 1. We have mentioned the same as quoted below in the manuscript, under the section titled “Exome-wide significant associations”.

“The study cohorts and phenotype definitions are described in Methods, and the cohort-specific sample sizes and participant demographics are summarized in Supplementary Tables 1 and 2 respectively.”

Figure 1b: Why are there no data from Sinai for *CHRNA2*?

We couldn’t define heavy smoking phenotype in the SINAI cohort as we didn’t have information about the number of cigarettes smoked per day or smoking pack years. However, we have reported *CHRNA2* pLOF plus missense burden association with ever-smoker phenotype in Supplementary figure 5b.

Figure 2. Panel b has not been described in the legend.

We've now added a description for panel b of Fig. 3 (previously Fig. 2).

Reviewer #2:

Rajagopal et al. performed an association study of multiple smoking phenotypes based on whole-exome sequencing data from multiple cohorts. They mainly focused on the rare variants in the coding regions and identified a signal in *CHRNA2* in the aggregation analyses. They also presented data on common variants in this region and other reported regions. They proposed that CHIP mutations explained another two signals in *ASXL1* and *DNMT3A*. This is an association study with large sample size. However, I have some concerns about the novelty of the results. In addition, some methods were not clearly described.

Originality and significance: if not novel, please include reference

The major finding of this study is a genetic signal in *CHRNA2*. However, the *CHRNA2* region has been reported in a previous GWAS (PMID: 30643251), as mentioned by the authors. I agree with the authors that the rare coding variants may contribute greatly to the susceptibility of phenotypes, but the novelty of the exome-wide screening will be greatly diminished if the region has been reported.

We highly value the past GWAS results and believe they can significantly help prioritize gene targets in the exome studies. But we disagree with the reviewer's view that the novelty of our finding is "greatly diminished" because of a prior report on the association of common variants near *CHRNA2*. We note that the GWAS locus near *CHRNA2* was among the >400 loci identified by Liu et al (2019)³, and the gene *CHRNA2* is one among the hundreds of genes mapped to the GWAS loci and reported in the supplementary tables. Furthermore, in the recent GWAS by Saunders et al (2022)¹, the *CHRNA2* locus is one among the >2000 loci and *CHRNA2* among >700 genes listed in the supplementary tables. Despite the well-known mechanistic links of *CHRNA2* with smoking addiction supported by numerous mouse studies, the gene name does not appear anywhere in the manuscript by Liu et al and Saunders et al, except in the supplementary table. That is in fact a pitfall of GWASs because they point to hundreds of genes and the most valuable findings get lost in the crowd. Rare variant studies such as ours help identify such important findings from the GWAS. In our analysis, we identified for the first-time significant associations between functional coding variants in *CHRNA2* with smoking behavior. This finding directly implicates *CHRNA2* as the causal gene and it also informs about the direction of the association, which corroborates with past animal studies^{4,5}. Even though the link between *CHRNA2* and smoking is itself not novel (it's been known since the 1990s), we emphasize that our finding demonstrating the protective effect of the loss of *CHRNA2* loss on smoking behavior in humans is indeed novel. If anything, our findings greatly complement the previous GWAS findings and highlight the importance of studying both GWAS and ExWAS findings together as we have discussed in the paper. So, we believe that neither the past GWASs diminish the value of our study nor ours undermine the value of past GWASs.

Only CHIP mutations of *ASXL1* and *DNMT3A* were associated with smoking behaviors; that is interesting. The authors propose their hypotheses (selective advantage); however, they do not

present additional supporting data. They should look at the variant allele fraction (VAF) of these mutations significantly associated with smoking phenotypes.

As per the reviewer's request, we have tested the associations of VAF in CHIP mutations in the eight most recurrent CHIP genes with the smoking phenotypes in a merged dataset of CHIP mutation carriers in the UKB (N=28,348) and GHS (N=11,063) cohorts. We aggregated the VAF estimates for CHIP mutations within each (and across all) of the eight genes and tested their associations with smoking phenotypes through regression analysis adjusted for age, sex, first 10 genetic PCs and a dummy variable for the cohort of origin. After correcting for multiple testing (FDR 1%), only the associations of VAF aggregated across all the genes with *ever-smoker* and *heavy-smoker* remained statistically significant. This is expected because the sample size drops when studying the VAF of individual genes. Looking at the nominally associated genes, the strongest association was seen for *ASXL1* VAF with *ever-smoker* and *heavy-smoker*. In addition, we also observed a few nominal associations for *TET2*, *TP53* and *SRSF2*. We report these results in Supplementary Table 10 and Supplementary Fig 11 (shown below). We added a short description of these results to the text under the title 'Associations of CHIP mutations in *ASXL1* and *DNMT3A*' in the results and added a paragraph in the methods (shown as tracked changes). And we have now deleted the text quoted below, as our confidence in the specificity of smoking associations with *ASXL1* and *DNMT3A* CHIP has reduced in the light of new nominal associations of smoking with VAF of *TET2* and other CHIP genes. We thank the reviewer for their suggestion to look at the VAF associations.

Deleted text:

“Notably, certain highly recurrent CHIP driver genes such as TET2 did not show significant associations with any of the six smoking phenotypes. This suggests that smoking influences the evolution of CHIP mutations through mechanisms that affect not all but only a specific set of genes”

Data & methodology: validity of approach, quality of data, quality of presentation

I do not see any methods for the analysis of CHIP mutations. How are these mutations defined? Also, I do not find the mean coverage of the whole-exome sequencing (not the percentage of 20X region); coverage might greatly influence the number of CHIP mutations. Is there any difference across different cohorts?

Detailed descriptions of CHIP mutation calling in the UKB and GHS cohorts and quality control procedures can be found in our recent publication⁶ focused exclusively on the genetics of CHIP. Since our primary focus in the current paper is not CHIP, we opted to mention the CHIP-related findings briefly and redirect interested readers to our primary CHIP paper. Since the reviewers have requested, we have now added a section in the method, titled “CHIP mutation analysis”, briefly describing the CHIP mutation calling, genetic and VAF association analysis of CHIP mutations (shown as track changes).

Regarding coverage, as part of the standard operating procedure, we always exclude samples with less than 80% coverage of target regions with at least 20x depth. And when evaluating the samples included in the final analysis, on average we capture greater than 95% of the target regions with at least 20x sequencing depth with often most of the participants (>98%) having more than 90% coverage. We use this measure because it provides a more accurate picture of genomic coverage than the mean. Cohort-specific sequencing metrics can be found in our primary exome-sequencing publications of the participating cohorts⁷⁻¹¹. The CHIP mutation associations are mainly driven by UK Biobank and GHS participants which contain older participants. We did not see any major differences in the CHIP calling between the two cohorts which we have investigated extensively in CHIP focused paper⁶.

The signal of *CHRNA2* was associated with the phenotype of heavy smoker in UKB/GHS/MCPS. The readers may also want to know the association between *CHRNA2* variants and the phenotype of ever smoker in the SINAI study.

We have reported the association of CHRN2 pLOF-plus-missense burden with *ever-smoker* in the SINAI cohort in Supplementary Fig. 5b.

The authors provide a single P value for analyses based on multiple cohorts. What is the combined strategy? Direct combination or meta-analysis? If direct combination, the author may need to describe how to correct the batch effects. If meta-analysis, they may need to show the results of reported signals in each cohort.

We performed ExWAS and GWAS within each of the cohorts separately and then meta-analyzed the results using an inverse-variance weighted approach using the METAL software. We have now revised the text in the methods under the section ‘Genetic association analysis’ accordingly as quoted below.

“Genetic association analyses were done within each of the cohorts separately using REGENIE software and the results were then meta-analyzed together using an inverse-variance weighted approach using METAL software. ...”

As per the reviewer’s request, we have now updated supplementary table 4 to show association statistics of all the significant variant and burden associations for individual cohorts and meta-analysis (in the first version of the manuscript only meta-analysis results were reported).

Page 7, line 23. The effect of CHRN4 pLOF is different from that in Supplementary Table 10. What is the mean of “Beta=0.65 SD”?

We apologize for the discrepancy between the values in the text and the table. The effect size in the text was based on conditional analysis, which was done to test if there is a statistically significant association beyond the known common variant signals. To avoid confusion, we now changed the effect sizes in the text to reflect the raw effect sizes from the ExWAS that were reported in the supplementary table. We revised the text as follows.

“Notably, the largest effect size was observed for the CHRN4 pLOF-only rare variant burden where the 13 pLOF carriers smoked on average ~6.8 fewer cigarettes per day more compared to non-carriers (Beta=0.68 SD; CI=0.17-1.18; P=0.008). This effect size is ~3 to 4 times larger than the largest effect size observed for CHRNA5 (Beta=0.23; CI=0.05-0.40; P=0.01) and CHRNA3 (Beta=0.16; CI=0.02-0.31; P=0.03) pLOF-only rare variant burden and ~7.5 times larger than rs16969968 (~1 cigarette more; Beta=0.09; CI=0.09-0.10; P=3.8e-125), a well-characterized common risk variant at this locus”

The quantitative phenotype cig-per-day was scaled to have a mean zero and standard deviation one to express the association effect sizes in terms of the number of SDs (one SD is approximately 10 cigarettes per day). So, an effect size of 0.68 SD would mean 6.8 cigarettes (0.68 x 10 cigarettes) difference between carriers and non-carriers.

Appropriate use of statistics and treatment of uncertainties

No statistical tests were performed in the ‘Interplay between common and rare variants’ section. Thus, any conclusions in this section may not be appropriate.

As per the reviewer's request, we have now added statistical analyses to this section. First, we analyzed the associations of PGS and *CHRNA2* rare variant burden with heavy smoking using a logistic regression model that included an interaction term between the two and appropriate covariates. We report the odds ratio, 95% confidence intervals and P value for the association of *CHRNA2* rare variant burden with heavy smoking (OR=0.66; 95% CI=0.56-0.79; P=3.4e-6) and the beta coefficient, standard error and P value for the association of PGS with heavy smoking (beta=0.33; SE=0.004; P=1e-300) and P value for the interaction term between the two (P=0.71), which was not significant as expected, corroborating past reports that show that the effects of rare and common variants are mostly independent of each other and additive. Second, we performed a quintile analysis where we divided the UKB participants into 5 equal groups and visualized the prevalence of heavy smoking in each of the groups separately in carriers and non-carriers of *CHRNA2* rare variant burden. To make a statistical comparison of the smoking prevalence between carriers and non-carriers within each quintile, we did logistic regression analysis within each group testing the association of *CHRNA2* rare variant burden with heavy smoking after adjusting for appropriate covariates. We report the odds ratio, 95% confidence interval and P value for each of the five groups in the main figure 4 (shown below). We have revised the manuscript sections accordingly shown as track changes (text under the title 'interplay between common and rare variants' in results and text under the title 'polygenic score analysis' in methods)

Page 5, line 33-34. No description of the methods for the enrichment analyses based on the data from the FinnGen project.

We have now added a section titled "FinnGen analysis" in the methods describing the enrichment analysis we performed in the FinnGen results as quoted below.

"We downloaded the associations of variant (rs202079239) with 3095 disease endpoints in the FinnGen database using their web browser (<https://r7.finnngen.fi/variant/1-154575801-C-G>). Through string search, we extracted associations related to smoking, substance abuse, addiction, COPD and other lung diseases. To test for enrichment of protective associations (OR<1) in the extracted phenotypes, we did a hypergeometric test using the 'phyper' function implemented in the R base package by passing the following values: q=36 (number of associations with OR<1 among the smoking-related phenotypes), m=2018(number of associations with OR<1 among all

the phenotypes), $n=1077$ (number of associations with $OR>1$ among all the phenotypes) and $k=47$ (total number of smoking-related phenotypes extracted).”

Conclusions: robustness, validity, reliability

Page 10, line 17-19. “However, to the best of our knowledge, what we describe here is the first human genetic evidence supporting the hypothesis that loss of *CHRNA2* protects against nicotine addiction.” This is not true. As mentioned by the authors themselves, Liu et al. have already presented the data of the *CHRNA2* region (rs2072659), as in Supplementary Table.

We have addressed this under comment 9. Liu et al³ only identified a common variant near *CHRNA2* and this does not conclusively prove that *CHRNA2* is the causal gene driving the association. It also does not provide information about the direction of the association, i.e., if increased or decreased gene function is linked to decreased smoking. On the other hand, our findings of protective associations between functional coding variants in *CHRNA2* with smoking directly implicate *CHRNA2* and support the hypothesis that loss of *CHRNA2* protects against nicotine addiction. Moreover, we have appropriately cited Liu et al (and now also, Saunders et al) and mentioned in the article that we identified the common variant signal searching through the results from Liu et al.

Suggested improvements: experiments, data for possible revision

Because smoking is the leading risk factor of lung cancer, the genetic variants associated with smoking commonly contribute to the risk of developing lung cancer (e.g., 15q25.1). Are rare *CHRNA2* variants associated with lung cancer risk in UKB data?

‘Lung cancer’ and ‘family history of lung cancer’ are among the many secondary phenotypes that we studied for associations with our main hits, and we have reported the results in Supplementary table 5. We did observe protective effect sizes for *CHRNA2* pLOF-plus-missense burden; however, they did not reach statistical significance (Lung cancer: $OR=0.85$; $CI=0.48-1.5$; $P=0.57$; family history of lung cancer: $OR=0.83$; $CI=0.69-1.0$; $P=0.06$).

In previous GWAS, more than 400 loci were reported to be associated with smoking. What about the associations for the rare variants in these loci?

We did investigate the rare variant associations at the GWAS loci mapped by Liu et al. and found no significant associations beyond *CHRNA2*. But we didn’t report it in our earlier draft. Motivated by the reviewer’s comment, we have now repeated this analysis using the latest GWAS results by Saunders et al¹. We studied the rare variant burden associations of two sets of genes: genes mapped to all the smoking-related GWAS loci and ‘high-priority genes’ i.e., genes mapped to GWAS loci with less than 5 fine-mapped variants as reported by Saunders et al. Unfortunately, as in the previous analysis, this new analysis did not find any significant associations (FDR 1%) beyond *CHRNA2*. We now report this in the manuscript. We have added a paragraph under the section ‘Association of rare variants at known GWAS loci’ and included a supplementary figure (shown below) visualizing the results.

Heavy-smokers were defined as who smoked 10 or more cigarettes per day. Additional sensitivity analysis should be performed to test the robustness of the definition, as this definition was different from other studies, which used pack-years of 20/30 to define heavy smokers.

We agree with the reviewer that the definition of heavy smoking varies from study to study. The reason why we chose the cut-off of ≥ 10 cigarettes per day is to align with the heavy-smoker phenotype in the MCPS which was defined as $\text{cig-per-day} \geq 10$. At the time of the analysis, we did not have information about cigarettes per day and so couldn't redefine the *heavy-smoker* phenotype in the MCPS cohort. As per the reviewer's request, we have now performed a sensitivity analysis testing the consistency of the protective association of *CHRNB2* pLOF-plus-missense burden across different definitions of *heavy-smoker*. We defined *heavy-smoker* in the UKB using four definitions: $\text{cig-per-day} \geq 10$ (the definition used in the paper), $\text{cig-per-day} \geq 20$, $\text{pack-years} \geq 20$ and $\text{pack-years} \geq 30$ and performed rare variant burden analysis for each of the definitions. As shown in the forest plot below, we observe significant protective associations with *CHRNB2* pLOF-plus-missense burden across all definitions.

CHRNB2.pLOF_plus_missense_burden				OR	CI	P value	MAF
Phenotype	Case counts	Control counts					
Heavy smoker(CPD \geq 10)	94,936 175 0	246,136 694 0		0.64	0.54,0.76	1.34e-07	0.001
Heavy smoker(CPD \geq 20)	56,959 101 0	246,136 694 0		0.61	0.49,0.75	1.75e-06	0.001
Heavy smoker(Pack years \geq 30)	38,715 66 0	246,136 694 0		0.59	0.46,0.77	3.57e-05	0.001
Heavy smoker(Pack years \geq 20)	65,284 111 0	246,136 694 0		0.58	0.47,0.72	7.59e-08	0.001

We have included this plot as supplementary figure 4 in the manuscript and added a line in the main results section as follows.

“The protective association of CHRNB pLOF-plus-missense burden with heavy smoking was observed irrespective of how we define heavy smoking (Supplementary Fig 4.)”

References: appropriate credit to previous work?

13. As whole-genome sequencing is gradually implemented in the population study (PMID: 32499645, 32581362, 33589841, 36113475), the authors may also discuss the limitation of the population study based on whole-exome sequencing.

Our major goal in this project is to identify potential drug targets by using a well-proven successful formula: WES at scale. We are still skeptical about the value of WGS in drug target discovery over WES. However, we now mention that using WES to study drug targets may miss noncoding variation as a limitation in the discussion.

“Finally, we have focused only on the coding regions of the genome captured via whole exome sequencing (WES) and so, we may have missed rare variants with large effects on smoking behavior residing in noncoding regulatory regions. With the recent increase in large-scale whole genome sequencing (WGS) efforts, rare large-effect regulatory variants influencing human diseases and traits are being discovered and such discoveries may have the potential to lead to drug targets. However, the question of whether WGS is a more cost-effective investment than WES for drug target discovery is yet to be answered.”

Clarity and context: lucidity of abstract/summary, appropriateness of abstract, introduction and conclusions

Abstract. “ $\alpha 4\beta 2$ is the predominant nAChR in human brain and is one of the targets of varenicline, a partial nAChR agonist/antagonist used to aid smoking cessation.” This is not the result of this association study. The description here is misleading.

We have now removed these lines from the abstract.

Reviewer #3:

The manuscript by Rajagopal and colleagues describes a comprehensive ExWAS of smoking for rare variants in up to 749,459 individuals across multiple ancestries. The study leveraged off several well-known cohorts including UK biobank, GHS, MCPS, and SINAI. They used 6 primary smoking phenotypes for their analysis. The main finding is that rare coding variants in CHRNA2 is protective for smoking. The authors also identified two other genes, ASXL1, DNMT3A that reached ExWAS significance and determined that these genetic signals were mainly due to the CHIP variants. The relationship between common, as well as PGS, and rare variants of the CHRNA2 with smoking were also explored.

There is a prior publication from the TOPMED study that described the contribution of rare variants to smoking based on WGS (Jang et al, PMID: 35927319). As the authors stated, that study was not able to pinpoint to any specific gene. This study is novel and add to our current knowledge of genetics associated with smoking behavior by analyzing rare variants.

Their findings also supported by animal-based studies which showed protective effects of CHRNA2 in nicotine dependence. Validation of a known 3'UTR SNP in CHRNA2 with this study also support the validity of CHRNA2 as a protective gene. This study supports CHRNA2 as a potential therapeutic target for smoking cessation and addiction and the availability of known drugs targeting this important class of proteins.

We thank the reviewer for summarizing and highlighting the major findings of our study.

Although the manuscript is well written, revision of the text to be more precise will improve the clarity and help the readers:

It probably better to move the supplemental fig 1 to the main text. The total N of subjects from each study included in this study should be indicated.

As per the reviewer's request, we have now moved the study design figure to main figure 1. Regarding sample size, we have provided cohort-specific and ancestry-specific sample sizes for the six smoking phenotypes in Supplementary Table 1.

Add total N to the "Case counts" and "Control counts" for Fig 1b for each gene.

In the forest plot, the Case counts and Control counts provide sample size broken down to wild type, heterozygotes and homozygotes for the minor allele. The total sum of these numbers will correspond to the total number of cases and controls which will be the same for all the genes and can be found in Supplementary Table 1.

The section "Associations of rare and common variants in CHRN2" in Result should be in two sections describing rare and common variants separately.

As per the reviewer's request, we have now split the rare and common variant associations in the results into two sections.

Long paragraph in the discussion maybe better in two paragraphs.

We have now split the second long paragraph of discussion into two.

References:

Saunders, G. R. B. *et al.* Genetic diversity fuels gene discovery for tobacco and alcohol use. *Nature*

612, 720–724 (2022).

Vink, J. M., Willemsen, G. & Boomsma, D. I. Heritability of Smoking Initiation and Nicotine Dependence. *Behav. Genet.* **35**, 397–406 (2005).

Liu, Mengzhen *et al.* Association studies of up to 1.2 million individuals yield new insights into the genetic etiology of tobacco and alcohol use. *Nat. Genet.* **51**, 237–244 (2019).

Picciotto, M. R. *et al.* Abnormal avoidance learning in mice lacking functional high-affinity nicotine receptor in the brain. *Nature* **374**, 65–67 (1995).

Picciotto, M. R. *et al.* Acetylcholine receptors containing the $\beta 2$ subunit are involved in the reinforcing properties of nicotine. *Nature* **391**, 173–177 (1998).

Kessler, M. D. *et al.* Common and rare variant associations with clonal haematopoiesis phenotypes.

Nature **612**, 301–309 (2022).

Backman, J. D. *et al.* Exome sequencing and analysis of 454,787 UK Biobank participants. *Nature*

599, 628–634 (2021).

Szustakowski, J. D. *et al.* Advancing human genetics research and drug discovery through exome sequencing of the UK Biobank. *Nat. Genet.* **53**, 942–948 (2021).

Van Hout, C. V. *et al.* Exome sequencing and characterization of 49,960 individuals in the UK Biobank. *Nature* **586**, 749–756 (2020).

Dewey, F. E. *et al.* Distribution and clinical impact of functional variants in 50,726 whole-exome sequences from the DiscovEHR study. *Science* **354**, aaf6814 (2016).

Ziyatdinov, A. *et al.* Genotyping, sequencing and analysis of 140,000 adults from the Mexico City Prospective Study. <http://biorxiv.org/lookup/doi/10.1101/2022.06.26.495014> (2022)
doi:10.1101/2022.06.26.495014.

Decision Letter, first revision:

31st Mar 2023

Dear Dr. Coppola,

Thank you for submitting your revised manuscript "Rare coding variants in CHRN2 reduce the likelihood of smoking" (NG-A61332R). It has now been seen by the original referees and their comments are below. The reviewers find that the paper has improved in revision, and therefore we'll be happy in principle to publish it in Nature Genetics, pending minor revisions to comply with our editorial and formatting guidelines.

Sincerely,
Wei

Wei Li, PhD
Senior Editor
Nature Genetics
New York, NY 10004, USA
www.nature.com/ng

Reviewer #1 (Remarks to the Author):

The authors have thoroughly responded to each of my (and the other reviewers') comments and have revised the manuscript accordingly. They included data from the newest GWAS to rerun analyses and they also performed additional analyses that were suggested by reviewer 2. I think this has improved the manuscript and I have no further comments.

Reviewer #2 (Remarks to the Author):

The authors have addressed all my concerns and improved their manuscript. I have no more questions.

Reviewer #3 (Remarks to the Author):

The authors have adequately addressed all of the comments and concerns from the reviewers.

Final Decision Letter:

4th May 2023

Dear Dr. Coppola,

I am delighted to say that your manuscript "Rare coding variants in CHRN2 reduce the likelihood of

smoking" has been accepted for publication in an upcoming issue of Nature Genetics.

Your paper will be published online after we receive your corrections and will appear in print in the next available issue. You can find out your date of online publication by contacting the Nature Press Office (press@nature.com) after sending your e-proof corrections. Now is the time to inform your Public Relations or Press Office about your paper, as they might be interested in promoting its publication. This will allow them time to prepare an accurate and satisfactory press release. Include your manuscript tracking number (NG-A61332R1) and the name of the journal, which they will need when they contact our Press Office.

Please note that *Nature Genetics* is a Transformative Journal (TJ). Authors may publish their research with us through the traditional subscription access route or make their paper immediately open access through payment of an article-processing charge (APC). Authors will not be required to make a final decision about access to their article until it has been accepted. [Find out more about Transformative Journals](https://www.springernature.com/gp/open-research/transformative-journals)

Authors may need to take specific actions to achieve [compliance](https://www.springernature.com/gp/open-research/funding/policy-compliance-faqs) with funder and institutional open access mandates. If your research is supported by a funder that requires immediate open access (e.g. according to [Plan S principles](https://www.springernature.com/gp/open-research/plan-s-compliance)) then you should select the gold OA route, and we will direct you to the compliant route where possible. For authors selecting the subscription publication route, the journal's standard licensing terms will need to be accepted, including [self-archiving-and-license-to-publish](https://www.nature.com/nature-portfolio/editorial-policies/self-archiving-and-license-to-publish). Those licensing terms will supersede any other terms

that the author or any third party may assert apply to any version of the manuscript.

Please note that Nature Portfolio offers an immediate open access option only for papers that were first submitted after 1 January, 2021.

If you have not already done so, we invite you to upload the step-by-step protocols used in this manuscript to the Protocols Exchange, part of our on-line web resource, natureprotocols.com. If you complete the upload by the time you receive your manuscript proofs, we can insert links in your article that lead directly to the protocol details. Your protocol will be made freely available upon publication of your paper. By participating in natureprotocols.com, you are enabling researchers to more readily reproduce or adapt the methodology you use. [Natureprotocols.com](https://natureprotocols.com) is fully searchable, providing your protocols and paper with increased utility and visibility. Please submit your protocol to <https://protocolexchange.researchsquare.com/>. After entering your [nature.com](https://www.nature.com) username and password you will need to enter your manuscript number (NG-A61332R1). Further information can be found at <https://www.nature.com/nature-portfolio/editorial-policies/reporting-standards#protocols>

Sincerely,
Wei

Wei Li, PhD
Senior Editor
Nature Genetics
New York, NY 10004, USA
www.nature.com/ng